# Orexin neurons inhibit sleep to promote arousal

Roberto De Luca 🄳 [1], Stefano Nardone[2,8], Kevin P. Grace[1,3,8], Anne Venner 🄳 [1], Michela Cristofolini[1], Sathyajit S. Bandaru[1], Lauren T. Sohn[1], Dong Kong[4], Takatoshi Mochizuki[5], Bianca Viberti[1], Lin Zhu[1], Antonino Zito 🄳 [6,7], Thomas E. Scammell[1], Clifford B. Saper 🄳 [1], Bradford B. Lowell[2], Patrick M. Fuller[1,3,9 ✉] & Elda Arrigoni[1,9 ✉]

Humans and animals lacking orexin neurons exhibit daytime sleepiness, sleep attacks, and state instability. While the circuit basis by which orexin neurons contribute to consolidated wakefulness remains unclear, existing models posit that orexin neurons provide their wake-stabilizing influence by exerting excitatory tone on other brain arousal nodes. Here we show using in vivo optogenetics, in vitro optogenetic-based circuit mapping, and single-cell transcriptomics that orexin neurons also contribute to arousal maintenance through indirect inhibition of sleep-promoting neurons of the ventrolateral preoptic nucleus. Activation of this subcortical circuit rapidly drives wakefulness from sleep by differentially modulating the activity of ventrolateral preoptic neurons. We further identify and characterize a feedforward circuit through which orexin (and co-released glutamate) acts to indirectly target and inhibit sleep-promoting ventrolateral preoptic neurons to produce arousal. This revealed circuitry provides an alternate framework for understanding how orexin neurons contribute to the maintenance of consolidated wakefulness and stabilize behavioral state.

[1] Department of Neurology, Division of Sleep Medicine, Beth Israel Deaconess Medical Center and Harvard Medical School, Boston, MA 02215, USA. [2] Department of Medicine, Division of Endocrinology, Diabetes and Metabolism. Beth Israel Deaconess Medical Center and Harvard Medical School, Boston, MA, USA. [3] Department of Neurological Surgery, University of California Davis School of Medicine, Davis, CA, USA. [4] Department of Pediatrics, Division of Endocrinology, F.M. Kirby Neurobiology Center. Children's Hospital and Harvard Medical School, Boston, MA, USA. [5] Department of Biology, Graduate School of Science and Engineering. University of Toyama, Toyama, Japan. [6] Department of Molecular Biology, Massachusetts General Hospital, Boston, MA 02114, USA. [7] Department of Genetics, Harvard Medical School, Boston, MA 02114, USA. [8] These authors contributed equally: Stefano Nardone, Kevin P. Grace. [10] These authors jointly supervised this work: Patrick M. Fuller, Elda Arrigoni. ✉email: pmfuller@ucdavis.edu; earrigon@bidmc.harvard.edu

Orexin (Ox; also called hypocretin) neurons in the lateral hypothalamus play an indispensable role in stabilizing and maintaining wakefulness. For example, loss of Ox-producing neurons in people with narcolepsy, results in debilitating excessive daytime sleepiness, sleep attacks, and wake-instability[1]. The narcoleptic phenotype is nearly fully recapitulated in Ox-deficient animal models[2–4]. While existing circuit models hold that Ox neurons stabilize wake by exciting other brain arousal nodes, —for example, Ox neurons increase wake by exciting noradrenergic locus coeruleus neurons[5], we hypothesized here that Ox neurons may instead (or also) produce their wake-promoting and wake-stabilizing effects through inhibition of neurons of the ventrolateral preoptic nucleus (VLPO), a region that is essential for initiating and maintaining sleep. The cellular VLPO contains neurons that are active during sleep, lesions of the VLPO result in profound and persisting insomnia, and acute stimulation of the VLPO, in particular resident GABA/galanin-containing VLPO neurons (VLPO$^{GABA/Gal}$), produces sleep[6–10]. Local administration of Ox into the VLPO also rouses animals from sleep, although how Ox acts on its postsynaptic cellular targets, and the identity of these targets, within the VLPO remains unresolved[11].

Given that Ox is an excitatory peptide, and that Ox neurons co-release glutamate, we predicted that Ox/glutamate input would not produce arousal by directly inhibiting sleep-promoting VLPO$^{GABA/Gal}$ neurons, as some other arousal inputs to the VLPO appear to do[12,13], but rather would inhibit VLPO$^{GABA/Gal}$ neurons indirectly. Ox might, for example, inhibit VLPO$^{GABA/Gal}$ neurons by activating a GABAergic afferent input to the VLPO, or do so via an additional, inhibitory synaptic relay. We hypothesized that Ox regulation of arousal depends upon a polysynaptic pathway from the lateral hypothalamus to an intra-VLPO circuit that comprises Ox-responsive GABAergic (VLPO$^{GABA}$) interneurons whose collaterals inhibit sleep-promoting VLPO$^{GABA/Gal}$ neurons to produce arousal.

To test our hypothesis, we first sought to determine if acute activation of presynaptic Ox inputs (Ox terminals) within the VLPO could trigger behavioral and electroencephalographic arousal from sleep. We then employed in vitro pharmacology, optogenetically-based circuit mapping, combined with single-cell transcriptomics, to determine the neurochemical and molecular phenotypes of the postsynaptic cellular targets of Ox input within the VLPO. In our final experimental step, we sought to identify and characterize the intra-VLPO circuitry and synaptic mechanisms through which Ox inputs to the VLPO might promote and support arousal. In this work, our findings identify a polysynaptic circuit, including an intra-VLPO inhibitory feed-forward circuit, through which Ox signaling can rapidly produce, and maintain, arousal.

## Results

### In vivo stimulation of orexin terminals in the VLPO rapidly triggers arousals from both NREM and REM sleep.

Orexin neurons innervate the preoptic region including the VLPO[14–16] (Fig. 1a). We first sought to determine whether activation of Ox terminals in the VLPO could produce or otherwise trigger arousal from sleep. To do so, we injected *AAV-DIO-ChR2-mCherry* or *AAV-DIO-ChR2-eYFP* into the Ox field of *Ox-IRES-Cre* mice and then optogenetically stimulated ChR2(+) Ox terminals in the VLPO in vivo, during both NREM and REM sleep. (Fig. 1). Selective expression of *mCherry* or *YFP* in Ox neurons was confirmed by double immunofluorescence labeling (Fig. 1d and Supplementary Table 1). On average, photostimulation of the Ox terminals in the VLPO consistently produced short-latency arousals during both NREM (Fig. 1e and g) and REM sleep (Fig. 1f and h), an effect not

seen in sham trials ($n = 5$; paired bootstrapping, $p > 0.05$). When stimulated during NREM sleep, we found that arousal probability in the ChR2-mCherry group ($n = 5$) was elevated above that in the wildtype group (WT; $n = 3$) for all stimulation levels (significant main effect of genotype $F_{(1, 24)} = 22.92$, $p = 0.003$; non-significant interaction between genotype and stimulation frequency $F_{(4, 24)} = 1.66$, $p = 0.192$, two-way ANOVA; Fig. 1i, *top*). However, the significant increases in arousal probability from NREM sleep were observed only during 10 and 20 Hz stimulations ($t = 2.98$ and $3.73$, $p = 0.019$ and $0.004$, respectively; *post-hoc* Holm-Šidák *t*-tests), not at the 1 and 5 Hz stimulation levels ($t = 0.20$ and $2.20$, $p = 0.842$ and $0.073$, respectively; *post-hoc* Holm-Šidák *t*-tests; $n = 5$). Comparison of the bootstrap confidence intervals and effect sizes at different stimulation frequencies revealed that the effect of 5 Hz stimulation was consistent with the effects of 10 and 20 Hz stimulations but inconsistent with the non-effect at the 1 Hz level (Fig. 1i, *bottom*). When stimulation was applied during REM sleep (interaction between genotype and stimulation frequency $F_{(2, 12)} = 7.02$, $p = 0.010$, two-way ANOVA one-repeated-measure; Fig. 1j, *top*), arousal probability increased significantly in association with both the 1 and 10 Hz stimulations ($t = 3.82$ and $5.16$, $p = 0.002$ and <0.001, respectively; sham versus 1 and 10 Hz conditions within the ChR2-mCherry group, *post-hoc* Holm-Šidák *t*-tests; $n = 5$; Fig. 1j, *bottom*). Interestingly, the arousal response during REM sleep appears to be specific to the Ox input to the VLPO given that similar optogenetic stimulation of GABAergic terminals, also arising from the lateral hypothalamic region, in the VLPO rouses mice from NREM but not REM sleep[13].

These results demonstrate that activation of Ox input to the VLPO produces rapid arousal from both NREM and REM sleep, and in a frequency-dependent manner. The arousals induced by photostimulation were typically brief. We also found that stimulation during NREM sleep, as compared with stimulations in REM sleep, doubled the length of the evoked arousals (Supplementary Fig. 1). Hence while activation of Ox terminals in the VLPO can evoke immediate arousal from NREM or REM sleep, only stimulation during NREM sleep was associated with longer-term effects, including an immediate post-activation suppression of sleep propensity. This finding might suggest that Ox input has downstream effects on separate NREM and REM sleep regulatory components, but these could both be located within the VLPO itself. This is consistent with previous pharmacological work showing that injections of Ox directly into the VLPO produces arousal by reducing both NREM and REM sleep and also confirming that Ox has local effects in the VLPO[11].

### Effects of orexin on the cellular VLPO neurons.

Hence, we next studied the effects of orexin-A (Ox-A) in VLPO neurons in brain slices of WT mice (Fig. 2a–c). We found a dual and opposite effect of Ox-A on VLPO neurons. Ox-A (0.3-1 μM) excited 35% of the neurons in the VLPO (firing frequency: $+173.29 \pm 41.29\%$; $n = 21$; resting membrane potential: $+2.80 \pm 0.44$ mV; $n = 26$; Fig. 2d), but unexpectedly inhibited 60% of the VLPO neurons (firing frequency: $-84.37 \pm 4.66\%$; $n = 37$; resting membrane potential: $-4.53 \pm 0.47$ mV; $n = 44$; Fig. 2e). Ox-A had no effects on the remaining 5% of neurons tested. We also found that while the Ox-A mediated inhibition was blocked by tetrodotoxin (TTX) or bicuculline, neither compound had an effect on Ox-A mediated excitation (Supplementary Fig. 2). These results demonstrate that while Ox *directly* excites some VLPO neurons, Ox also inhibits other VLPO neurons, but this inhibition must occur *indirectly* and, we predict, involves synaptic GABA signaling. We also found that the VLPO neurons that were excited by Ox-A were also excited by the cholinergic agonist carbachol and that

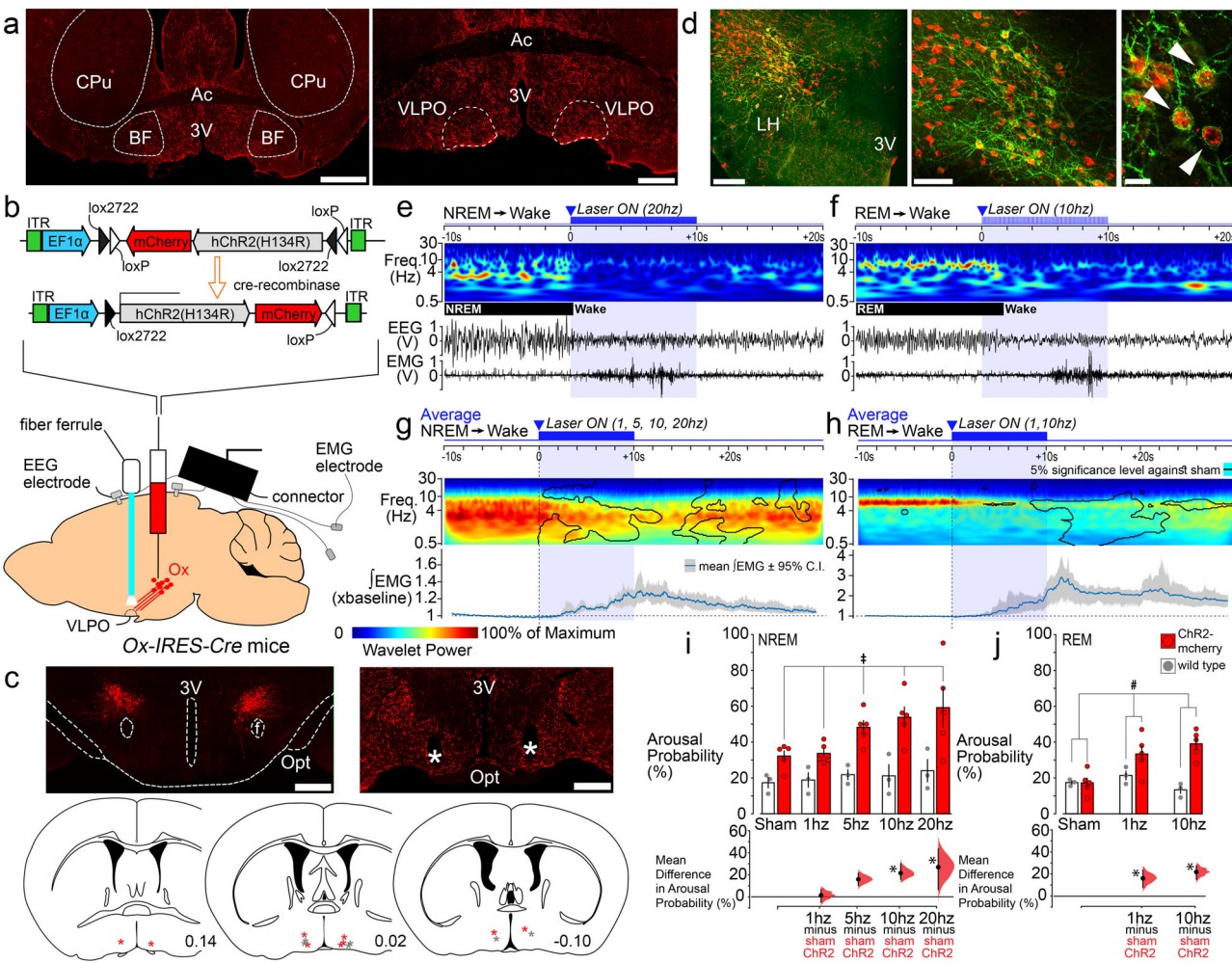

**Fig. 1 Optogenetic stimulation of orexin terminals in the VLPO rapidly triggers arousals from both NREM and REM sleep. a** Orexin innervation of the POA and VLPO (scale bars: 1 and 0.5 mm). **b** *AAV-DIO-ChR2-mCherry* was injected bilaterally into the Ox field of *Ox-IRES-Cre* mice and optical fibers bilaterally placed above the VLPO. **c** Orexin neurons transduced with ChR2- mCherry (red; left) and optical fiber (*) above the VLPO (right; scale bars: 500 μm). Optical fiber placements (red * ChR2-mCherry and grey * WT; bottom). **d** ChR2-eYFP(+) neurons double-labeled for Ox-A (in red; scale bars: 200; 100, and 20 μm). Laser-induced arousals from NREM (**e**) and REM (**f**) sleep (10 s-long periods before and after 10 s light pulses, top). EMG and EEG recordings and continuous EEG wavelet transforms (bottom). Trial-averaged responses to laser stimulation in NREM (**g**) and REM (**h**) sleep. EEG wavelet power for trials containing arousals (top; $n = 5$). Black contour lines: 5% significance level against sham trials (paired bootstrap confidence interval of the mean wavelet power difference). EMG mean responses (grey region: 95% bootstrap confidence interval; bottom). Mean differences in arousal probability, from NREM (**i**) and REM sleep (**j**) in ChR2-mCherry ($n = 5$) and WT ($n = 3$; top). On the bottom, the mean differences in arousal probability (stimulation vs. sham) within the ChR2-mCherry group (NREM: effect of genotype $F_{(1, 24)} = 22.92$, $p = 0.003$; interaction between genotype and stimulation frequency $F_{(4, 24)} = 1.66$, $p = 0.192$, two-way ANOVA one-repeated-measure. REM: interaction between genotype and stimulation frequency $F_{(2, 12)} = 7.02$, $p = 0.010$, two-way ANOVA one-repeated-measure). Mean difference in arousal probability from NREM (**i**, bottom; $t = 2.98$ and $3.73$, $p = 0.019$ and $0.004$ for 10 and 20 Hz stimulations) and REM sleep (**j**, bottom; $t = 3.82$ and $5.16$, $p = 0.002$ and $0.0007$ for 1 and 10 Hz stimulations; post-hoc Holm–Šidák $t$-tests, two-way ANOVA one-repeated-measure; $n = 5$; Source Data file). Error bars: SEM; ‡: significant for factor genotype (two-way ANOVA); #, significant interaction between the factors genotype and stimulation frequency (two-way ANOVA). Mean differences (stimulations vs sham within the ChR2-mcherry group) plotted as bootstrap sampling distributions (bottom). Dots: mean difference in arousal probability; vertical error bars: 95% confidence intervals; *$p < 0.05$ post-hoc Holm–Šidák $t$-test. All testing was two-tailed, and $n$ refers to the number of independent animals. 3V 3rd ventricle, Opt optic tract, f fornix, Ac anterior commissure, CPu corpus striatum, BF basal forebrain and LH lateral hypothalamus. Atlas levels are per Franklin and Paxinos, 2001.

the VLPO neurons that were inhibited by Ox-A were, instead, inhibited by carbachol (Supplementary Fig. 3). These findings suggest that the VLPO contains at least two distinct neuronal populations that respond in opposite manners to wake- promoting signals, suggesting that these two subpopulations differentially regulate sleep and arousal.

**VLPO GABAergic neurons show a dual response to orexin.** The VLPO is largely composed of GABAergic neurons,

including a subpopulation that express the peptide galanin (VLPO$^{GABA/Gal}$). VLPO$^{GABA/Gal}$ neurons are active during sleep and promote NREM sleep[6,9,17]. Currently, little is known about the role in sleep-wake regulation of VLPO GABAergic neurons that lack galanin (VLPO$^{GABA}$). To study these neurons, we first focused on the effects of orexin (Ox-A) on the VLPO$^{GABA}$ population. We recorded from fluorescently labeled VLPO$^{GABA}$ neurons using *Vgat-IRES-Cre* mice into which we had placed microinjections of *AAV-DIO-TdTomato* or *AAV-*

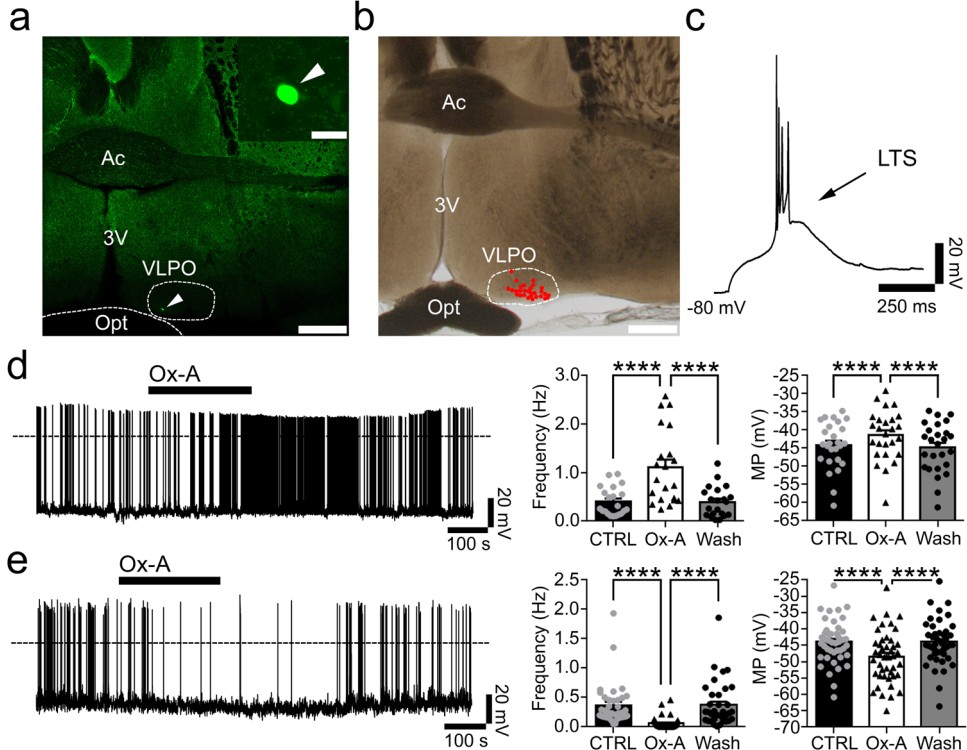

**Fig. 2 Dual response of orexin in VLPO. a** Whole-cell recordings of VLPO neurons and *post-hoc* labeling of a biocytin-filled neuron (scale bars: 500 and 20 μm). **b** Map of the recorded neurons in VLPO ($n = 29$; scale bar: 500 μm). **c** Most of VLPO neurons exhibit LTS (80 out of 116). **d** Ox-A (0.3-1 μM) excites 35% of the VLPO neurons: it increases action potential firing frequency ($n = 21$; one-way ANOVA, $F_{(2, 60)} = 30.49$; $p < 0.0001$; CTRL vs Ox-A and Ox-A vs. Wash, *adj-p* $< 0.0001$) and depolarizes the membrane potential ($n = 26$; one-way ANOVA, $F_{(2, 75)} = 49.89$; $p < 0.0001$; CTRL vs Ox-A and Ox-A vs Wash, *adj-p* $< 0.0001$). **e** Ox-A inhibits 60% of the VLPO neurons: it reduces action potential firing frequency ($n = 37$; one-way ANOVA, $F_{(2, 108)} = 28.56$; $p < 0.0001$; CTRL vs Ox-A and Ox-A vs Wash, *adj-p* $< 0.0001$) and hyperpolarizes membrane potential ($n = 44$; one-way ANOVA, $F_{(2, 129)} = 63.18$; $p < 0.0001$; CTRL vs Ox-A and Ox-A vs Wash, *adj-p* $< 0.0001$). ****$p < 0.0001$, Bonferroni's *post-hoc* test. Ac, anterior commissure; 3V, 3rd ventricle; Opt, optical chiasm. Panel **d** and **e**: data are represented as means ± SEM, *n* refers to the number of recorded neurons and Source Data are provided as a Source Data file.

*DIO-GFP* into the VLPO (Supplementary Fig. 4a,b). Both AAVs showed selective expression of fluorescent protein in Vgat-expressing neurons (Supplementary Fig. 5 and Supplementary Table 1). We found that Ox-A excited 42.8% of VLPO[GABA] neurons (firing frequency: $+225.61 \pm 95.01\%$; $n = 6$ and resting membrane potential: $+2.95 \pm 1.06$ mV; $n = 6$) and inhibited 50% of VLPO[GABA] neurons (firing frequency: $-72.92 \pm 14.26\%$; $n = 7$ and resting membrane potential: $-3.40 \pm 1.13$ mV; $n = 7$). Ox-A was without effect on the remaining 7% of the VLPO[GABA] neurons. These results indicate that, within the VLPO, and similar to our initial findings, there are at least two populations of VLPO[GABA] neurons, one that is excited by Ox and one that is instead inhibited.

Previous studies showed that the VLPO[GABA/Gal] sleep-promoting neurons are inhibited by noradrenaline (NA)[12,18]. Accordingly, we found that the VLPO[GABA] neurons that were inhibited by Ox-A were also inhibited by NA ($n = 12$) and that they expressed Gad1, Gad2, and galanin, indicating these are the VLPO[GABA/Gal] neurons. In contrast, we found that VLPO[GABA] neurons that were excited by Ox-A were also excited by NA ($n = 24$) and that they expressed Gad1, and Gad2, but not galanin, i.e., were VLPO[GABA] neurons (Fig. 3 and Supplementary Fig. 4). These results confirm and extend previous studies demonstrating that the VLPO contains at least two subgroups of GABAergic neurons. The first of these subgroups, we predict, are VLPO[GABA/Gal] neurons, which express galanin, are inhibited by wake-promoting signals such as NA and Ox, and comprise the subpopulation of VLPO neurons that promote sleep[9,12,17]. The second subgroup, VLPO[GABA] neurons, lack galanin, are excited by NA and Ox and, we also predict, could be responsible for Ox-mediated feedforward inhibition of VLPO[GABA/Gal] neurons that would drive arousal.

**Orexin inhibits VLPO[GABA/Gal] neurons by increasing GABAergic afferent input.** Our next experiment focused on sleep-promoting VLPO[GABA/Gal] neurons. We recorded from TdTomato-labelled VLPO[GABA/Gal] neurons in *Gal-IRES-Cre* mice that had received injections of *AAV-DIO-TdTomato* into the VLPO. TdTomato-labeled VLPO[GABA/Gal] neurons expressed GABAergic markers, and as predicted, they were inhibited by NA. The majority of VLPO[GABA/Gal] neurons exhibited low-threshold spikes (LTS) (88.5%; $n = 26$), and importantly, they were inhibited by Ox-A (76.5%; action potential frequency: $-84.40 \pm 6.05\%$; $n = 12$; and membrane potential: $-4.60 \pm 0.93$ mV; $n = 13$; Supplementary Fig. 6). Given again that Ox is an excitatory peptide, the finding that Ox-A inhibits VLPO neurons suggests that the observed inhibition by Ox-A must be an indirect effect and, moreover, might imply at least one additional synaptic relay and possibly the involvement of synaptic GABA. We, therefore, tested the effects of Ox-A on GABAergic synaptic input to VLPO[GABA/Gal] neurons. Ox-A increased the frequency of spontaneous inhibitory postsynaptic currents (sIPSCs) in 75% of the VLPO[GABA/Gal] neurons ($+58.76 \pm 13.29\%$; $n = 9$), indicating that

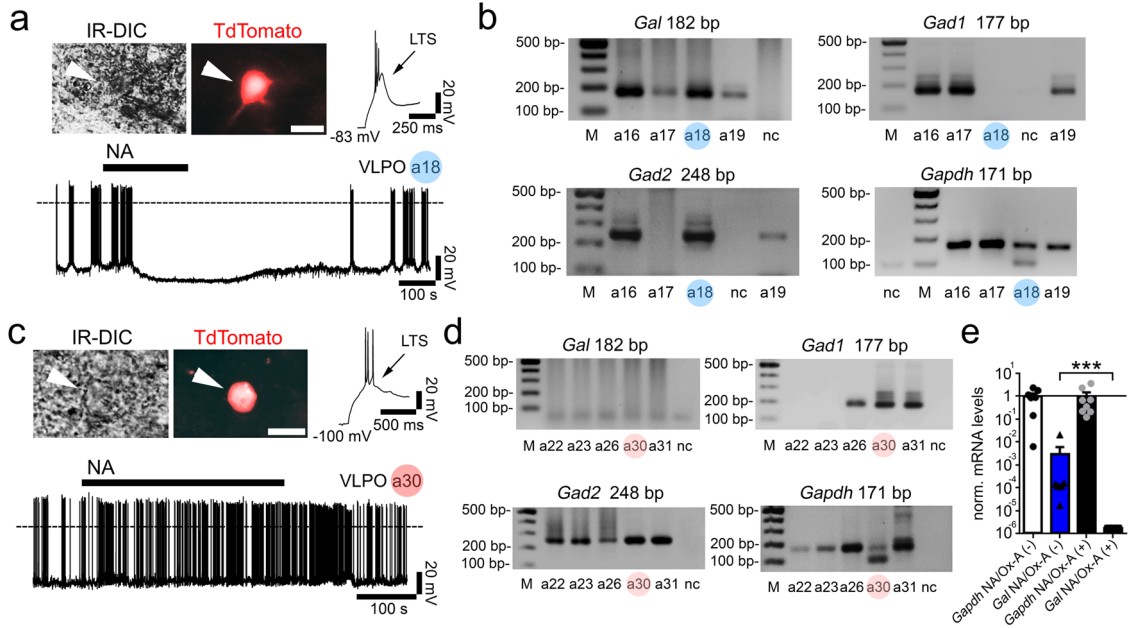

**Fig. 3 Pharmacological and molecular diversity of the VLPO GABAergic neurons.** We recorded from TdTomato labeled VLPO Vgat(+) (VLPO$^{Vgat}$) neurons in *Vgat-IRES-Cre* mice injected into the VLPO with *AAV-DIO-TdTomato*. VLPO$^{Vgat}$ neurons (**a** *top;* TdTomato-labelled; sample #a18; scale bar: 20 μm) that are inhibited by NA (50 μM) express galanin. Single-cell RT-PCR results from 4 VLPO neurons inhibited by NA (**b**). The VLPO$^{Vgat}$ neurons (**c** *top;* TdTomato-labelled; sample #a30; scale bar: 20 μm) that are excited by NA do not express galanin. Single-cell RT-PCR results from 5 VLPO$^{Vgat}$ neurons excited by NA (**d**). *Gal* (182 bp), *Gad1* (177 bp), *Gad2* (248 bp), and *Gapdh* (171 bp; housekeeping gene), M marker ladder, nc negative control. **e** scRT-sqPCR results confirming the presence of *galanin* mRNA in 7 VLPO$^{Vgat}$ neurons inhibited by NA and/or Ox-A (NA/Ox-A(−)) and the absence of *galanin* mRNA in 8 VLPO$^{Vgat}$ neurons excited by NA and/or Ox-A (NA/Ox-A(+)). Mann–Whitney unpaired one-sided *t*-test, *Gal* NA/Ox-A(−) vs *Gal* NA/Ox-A(+); $p = 0.0002$, ****p* < 0.001; values normalized to the mean *Gapdh*. **e** data are represented as means ± SEM. Panels **b**, **d**, and **e** Source Data are provided as a Source Data file.

Ox likely inhibits VLPO$^{GABA/Gal}$ neurons by increasing the GABAergic afferent input (Supplementary Fig. 7).

**Orexin mediates feedforward inhibition of VLPO$^{GABA/Gal}$ neurons via VLPO$^{GABA}$ neurons.** Given that most VLPO$^{GABA}$ neurons are directly excited by Ox, we next sought to determine whether VLPO$^{GABA}$ neurons could mediate feedforward inhibition of VLPO$^{GABA/Gal}$ neurons. We first tested whether VLPO$^{GABA}$ neurons inhibit VLPO$^{GABA/Gal}$ neurons (VLPO$^{GABA}$ → VLPO$^{GABA/Gal}$ input) and then whether this input was positively regulated by Ox (Fig. 4). Using an intersectional approach, we placed microinjections of Flp-dependent *AAV-fDIO-ChR2-eYFP* and Cre-dependent *AAV-DIO-TdTomato* into the VLPO of *Vgat-Flp::Gal-IRES-Cre* mice. This approach produced expression of ChR2 in VLPO neurons that express the vesicular GABA transporter (Vgat; VLPO$^{Vgat}$) and *TdTomato* in VLPO neurons that express galanin. RNA scope in situ hybridization confirmed selective expression of *YFP* and *TdTomato* in VLPO$^{Vgat}$ and VLPO$^{GABA/Gal}$ neurons, respectively (Supplementary Fig. 8 and Supplementary Table 1). Histological assessment of the AAV injections further confirmed that ChR2 expression was restricted to the VLPO region (Fig. 4b). We recorded from the fluorescently labeled VLPO$^{GABA/Gal}$ neurons at the reversal potential of the ChR2-mediated current[19,20] while photostimulating VLPO$^{Vgat}$ neurons. Photostimulation of the VLPO$^{Vgat}$ neurons produced short-latency opto-evoked IPSCs (oIPSCs) in VLPO$^{GABA/Gal}$ neurons (Fig. 4).

As both VLPO$^{GABA}$ and VLPO$^{GABA/Gal}$ neurons express *Vgat* we then tested whether collateral projections from VLPO$^{GABA/Gal}$ neurons (VLPO$^{GABA/Gal}$ → VLPO$^{GABA/Gal}$) could contribute to the oIPSCs evoked in the VLPO$^{GABA/Gal}$ neurons when photo-stimulating the VLPO$^{GABA}$ neurons. To explore this possibility, we

placed injections of *AAV-DIO-ChR2-mCherry* into the VLPO of *Gal-IRES-Cre* mice and recorded from mCherry-labeled VLPO$^{GABA/Gal}$ neurons. While photostimulation of VLPO$^{Vgat}$ input triggered oIPSCs in 92.7% ($n = 55$) of VLPO$^{GABA/Gal}$ neurons, photostimulation of VLPO$^{GABA/Gal}$ input failed to produce oIPSCs in any of the VLPO$^{GABA/Gal}$ neurons tested ($n = 6$; Fig. 4c,d). These results demonstrate an absence of reciprocal connectivity between VLPO$^{GABA/Gal}$ neurons, and more importantly show that activation of VLPO$^{Vgat}$ neurons by Ox is the mechanism by which the VLPO$^{GABA}$ → VLPO$^{GABA/Gal}$ circuit is activated.

Photostimulation of VLPO$^{GABA}$ neurons produced GABA$_A$-mediated oIPSCs in VLPO$^{GABA/Gal}$ neurons ($n = 8$; Fig. 4e,f). Furthermore, the oIPSCs persisted in the presence of TTX (1 μM + 4-aminopyridine, 4-AP 25–50 μM) indicating monosynaptic connectivity ($n = 43$; Fig. 4g). Ox-A increased the amplitude of oIPSCs recorded in TTX (+59.71 ± 26.3%) in 56% of VLPO$^{GABA/Gal}$ neurons ($n = 16$; Fig. 4h,i) indicating a presynaptic effect. All together these results demonstrate that VLPO$^{GABA/Gal}$ neurons can be inhibited by local VLPO$^{GABA}$ neurons, and that activity of this circuit is enhanced by Ox.

We next applied peak-scaled non-stationary fluctuation analysis[21–23] to assess whether Ox-A enhances the VLPO$^{GABA}$ → VLPO$^{GABA/Gal}$ input by increasing the unitary GABA$_A$ current or the number of activated GABA$_A$ receptors or both. For each cell, we obtained parabolic variance vs current amplitude curves. We estimated the ionic channel current ($i$) and the number of activated channels ($N$) open at the peak of the oIPSCs, in control and during Ox-A application. We found that Ox-A increased the oIPSC amplitude by enhancing the number of activated GABA$_A$ channels (+89.33 ± 33.23 %; $n = 8$, $p = 0.0091$, paired *t*-test) without affecting the GABA$_A$ unitary current (Fig. 4j). These

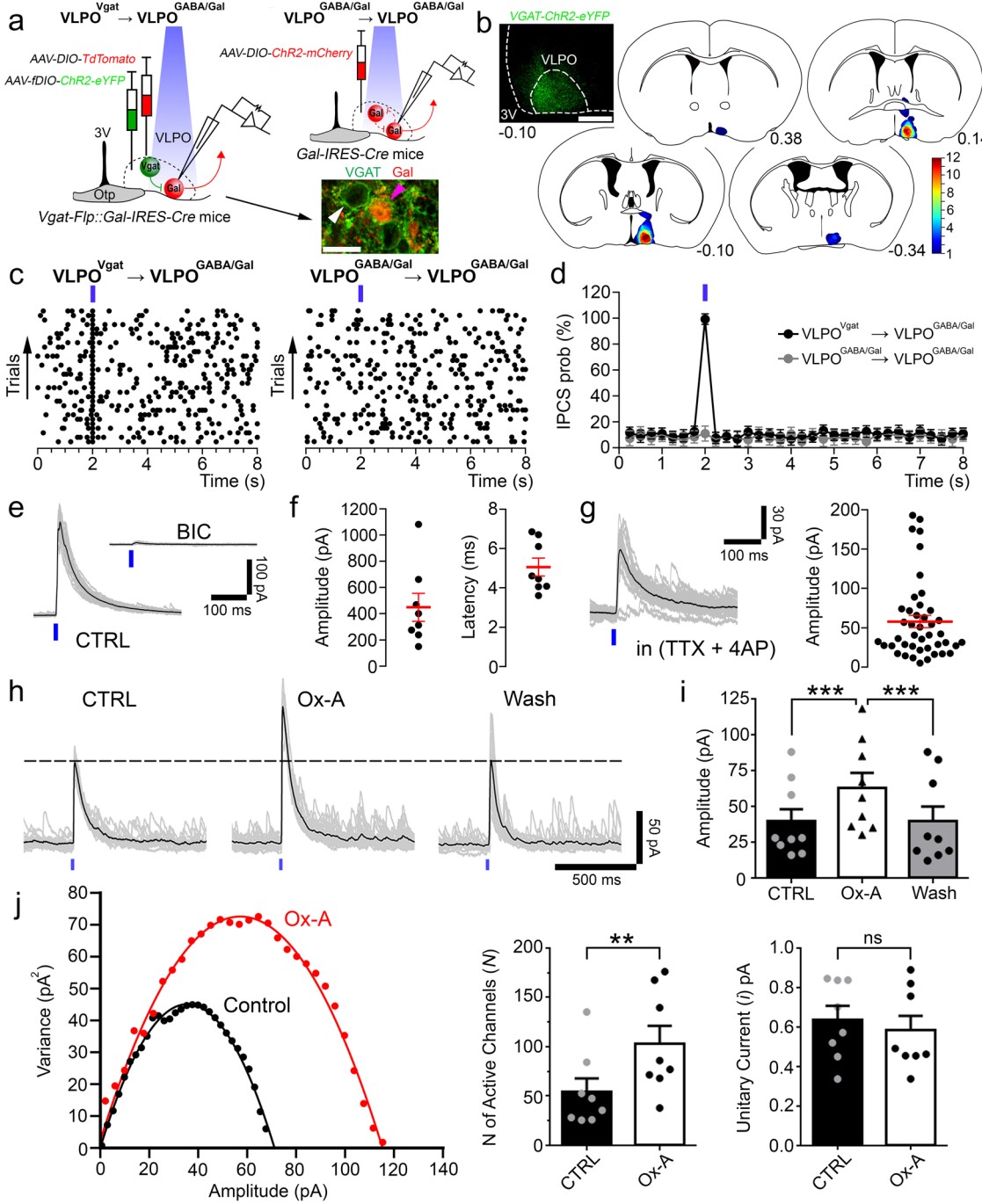

results are a strong indication that Ox enhances the VLPO-$^{GABA}$ → VLPO$^{GABA/Gal}$ input by increasing the amount of GABA being released.

**VLPO$^{GABA/Gal}$ and VLPO$^{GABA}$ neurons have distinct molecular profiles and expression of orexin receptors is restricted to a subset of VLPO$^{GABA}$ neurons.** To characterize the transcriptional profile of the VLPO$^{GABA/Gal}$ and VLPO$^{GABA}$ neurons, we retrieved and clustered the publicly available data of the preoptic area (POA) previously reported by Moffitt and colleagues in 2018[24]. Our bioinformatic analysis confirmed their earlier published POA clusters along with the cellular markers (Supplementary Fig. 9, Supplementary Table 2). Using the spatial

transcriptomic (*Multiplexed Error Robust Fluorescence In Situ Hybridization*: MERFISH) data provided in the same study, we isolated the neuronal clusters of the VLPO region and generated a VLPO atlas of 4174 neurons x 21,043 genes divided into 20 (#0-19) clusters (Supplementary Fig. 10,11, Supplementary Table 3). We then compared the molecular identity of GABAergic neuronal groups in VLPO that either express or lack expression of galanin, by performing differential gene expression. The VLPO$^{GABA/Gal}$ and VLPO$^{GABA}$ groups comprised 3 clusters (515 neurons) and 12 clusters (2,911 neurons), respectively. We found that the VLPO-$^{GABA/Gal}$ and VLPO$^{GABA}$ groups have very distinct transcriptional profiles (Fig. 5a-b and Supplementary Table 4), but also share commonalities. For example, both cell populations expressed

**Fig. 4 Orexin enhances the VLPO$^{GABA}$→VLPO$^{GABA/Gal}$ circuit. a** To study the VLPO$^{Vgat}$→VLPO$^{GABA/Gal}$ input, we injected a mix of *AAV-fDIO-ChR2-eYFP* and *AAV-DIO-TdTomato* into the VLPO of *Vgat-Flp::Gal-IRES-Cre* mice ($n = 10$) and recorded from TdTomato-labeled VLPO$^{GABA/Gal}$ neurons. To study the VLPO$^{GABA/Gal}$→VLPO$^{GABA/Gal}$ input, we injected *AAV-DIO-ChR2-mCherry* into the VLPO of *Gal-IRES-Cre* mice ($n = 2$) and recorded from mCherry-labeled VLPO$^{GABA/Gal}$ neurons. VLPO$^{GABA}$ neurons labelled in green (white arrow) and VLPO$^{GABA/Gal}$ neurons double-labeled in green and red (magenta arrow; scale bar: 20 μm). **b** Restricted transduction of *AAV-fDIO-ChR2-eYFP* in the VLPO of *Vgat-Flp::Gal-IRES-Cre* mice (YFP immunolabeling in green, scale bar: 500 μm). Injection sites from 12 mice used for in vitro CRACM recordings. **c** Raster plots of IPSCs in VLPO$^{GABA/Gal}$ neurons with photostimulation of VLPO$^{Vgat}$→VLPO$^{GABA/Gal}$ input (*left*) and VLPO$^{GABA/Gal}$→VLPO$^{GABA/Gal}$ input (*right*; bin duration: 50 ms). **d** IPSC probability in response to photostimulation of the VLPO$^{Vgat}$→VLPO$^{GABA/Gal}$ input (*black*; $n = 8$) and the VLPO$^{GABA/Gal}$→VLPO$^{GABA/Gal}$ input (*grey*; $n = 6$; means ± SEM). Photostimulation of VLPO$^{Vgat}$ neurons evokes GABA$_A$-mediated oIPSCs in VLPO$^{GABA/Gal}$ neurons (**e**; BIC 20 μM). Mean oIPSC amplitude and latency (**f**; $n = 8$; in red: means ± SEM). **g** Opto-evoked IPSCs recorded in TTX (1 μM + 4-AP 25–50 μM, $n = 43$; in red: mean ± SEM). **h** Ox-A (1 μM) increases oIPSC amplitude (recordings in TTX + 4-AP). **i** Mean oIPSC amplitude in CTRL, Ox-A and Wash ($n = 9$, one-way ANOVA), $F_{(2, 24)} = 19.42$, $p < 0.0001$, CTRL vs Ox-A and Ox-A vs Wash adj-*p*-value = 0.0002; means ± SEM). **j** Peak-scaled non-stationary fluctuation analysis showing the current/variance relationship for oIPSCs (left; control: $N = 84.2$, $i = 0.84$ pA and Ox-A: $N = 139.6$, $i = 0.82$ pA). Ox-A increases the numbers of activated GABA$_A$ channels (*center*; $n = 8$, CTRL vs Ox-A, $p = 0.0091$, *paired one-sided t*-test; **$p < 0.01$) without affecting the GABA$_A$ unitary current (*right*; $n = 8$, CTRL vs Ox-A, $p = 0.2052$, paired one-sided *t*-test; means ± SEM). Recordings at reversal potential of the ChR2-mediated current (Vh = −5 to −15 mV). Blue-light pulses (10 ms; blue bars). ***$p < 0.001$ Bonferroni's *post-hoc* test. In grey: 30 individual oIPSCs; in black: average oIPSC. Opt, optical chiasm; 3V, 3$^{rd}$ ventricle; Atlas levels are per Franklin and Paxinos, 2001. For panels **d**–**j**: *n* refers to the number of recorded neurons. Panels **c**, **d**, **f**, **g**, and **i**, **j**: Source Data are provided as a Source Data file.

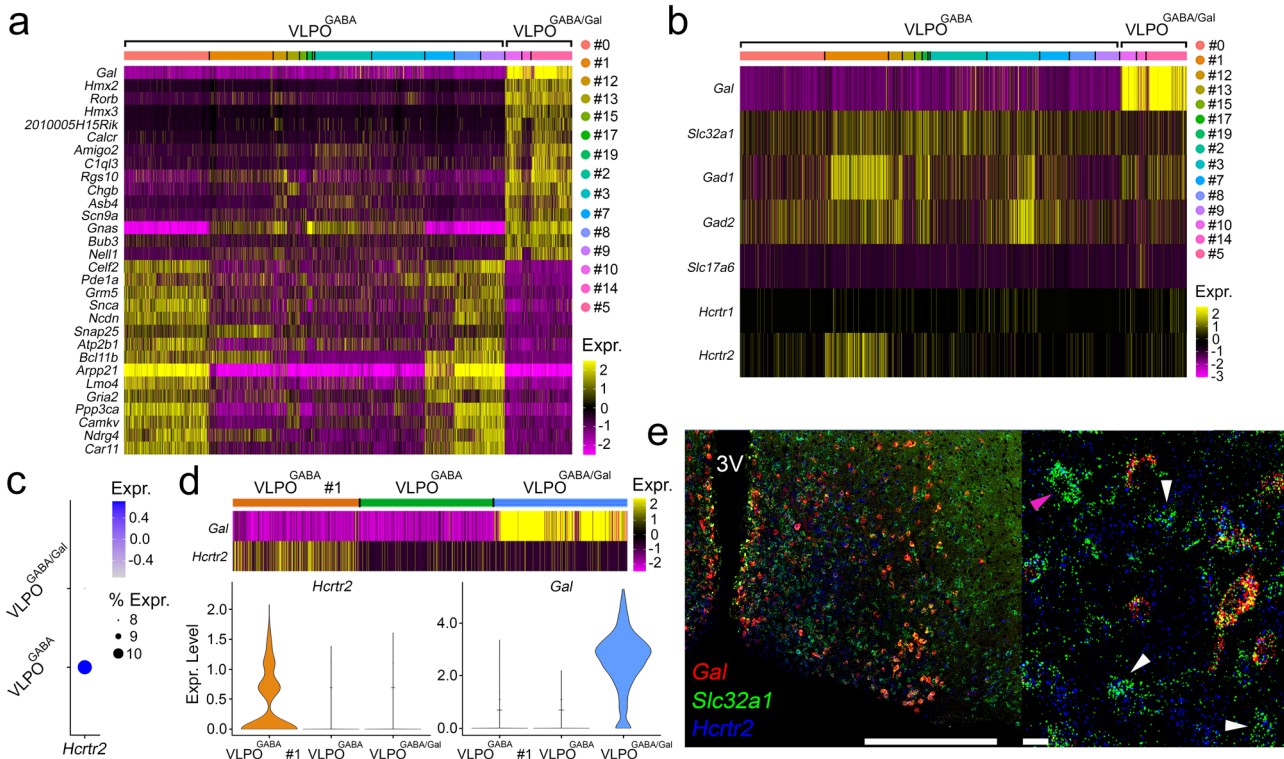

**Fig. 5 Differential expression profiles and orexin receptor expression in VLPO$^{GABA/Gal}$ and VLPO$^{GABA}$ neurons.** Heat maps representing the expression levels of the top 30 differentially expressed genes ranked by *adj-p* (panel **a**; top 15 hypo- and top 15 hyper-expressed) and expression levels of GABAergic genes (*Slc32a1, Gad1, Gad2*), *Hcrtr1, Hcrtr2, Gal* and *Slc17a6* (panel **b**) in the VLPO$^{GABA}$ (#0–3; 7–9; 12, 13, 15, 17, 19) and VLPO$^{GABA/Gal}$ (#5, 10, 14) clusters. *Hcrtr2* positive neurons were largely restricted to VLPO$^{GABA}$ cluster #1. **c** Dot plot representing the average expression of *Hcrtr2* gene (*x axis*) in VLPO$^{GABA}$ and VLPO$^{GABA/Gal}$ groups (*y axis*). Dot size: percentage of neurons that expresses a specific gene (*bottom right*). Color intensity: expression level (*top right*). **d** Heat map of *Gal* and *Hcrtr2* gene expression levels in VLPO$^{GABA}$ cluster #1 (orange; 489 neurons), in all the other VLPO$^{GABA}$ clusters (#0, 2, 3; 7–9; 12, 13, 15, 17, 19; green; down sampled to 515 neurons) and merged VLPO$^{GABA/Gal}$ clusters (#5, 10, 14; cyan; 515 neurons). Violin plots for *Hcrtr2* (*bottom left*) and *Gal* (*bottom right*) differential gene expression: VLPO$^{GABA}$ cluster #1 (orange), all the other VLPO$^{GABA}$ clusters (#0, 2, 3; 7–9; 12, 13, 15, 17, 19; green) and VLPO$^{GABA/Gal}$ clusters (#5, 10, 14; cyan). *Hcrtr2* is expressed in VLPO$^{GABA}$ #1 (VLPO$^{GABA}$ #1 vs VLPO$^{GABA/Gal}$ adj-*p* value = 4.98E−32; VLPO$^{GABA}$ #1 vs. VLPO$^{GABA}$ (#0, 2, 3; 7–9; 12, 13, 15, 17, 19) adj-*p* value = 1.63E−114). *Gal* is expressed in VLPO$^{GABA/Gal}$ neurons (VLPO$^{GABA}$ #1 vs VLPO$^{GABA/Gal}$ adj-*p* value = 1.26E−144; VLPO$^{GABA}$ (#0, 2, 3; 7–9; 12, 13, 15, 17, 19) vs VLPO$^{GABA/Gal}$ adj-*p* value = 0). Heat map and Dot plot expression values are represented as z-scores. Expression levels is color-coded in the legend. In the heat maps, *x-axis*: individual clusters color-coded bar; *y-axis*: genes. As color dots: Cluster IDs (*right*). Test used: *Wilcoxon Rank Sum two-sided* Bonferroni-corrected Test. **e** Expression in VLPO of *Gal* (red), *Slc32a1* (green) and *Hcrtr2* (pseudo colored in blue) mRNAs. White arrows: *Slc32a1*(+), *Hcrtr2*(+) and *Gal*(−) neurons; magenta arrows: *Slc32a1*(+), *Hcrtr2*(−) and *Gal*(−) neurons. Scale bars: 500 and 20 μm (*bottom right*). 3V, 3$^{rd}$ ventricle. Source Data are provided as a Source Data file.

markers for GABA synthesis and transmission (*Slc32a1*, *Gad1*, *Gad2*) and neither expressed the *Vglut2* gene (*Slc17a6*). In agreement with our electrophysiology data, orexin/hypocretin receptors (Hcrtrs) were not expressed in any of the three VLPO$^{GABA/Gal}$ clusters, whereas cluster #1 of the VLPO$^{GABA}$ group (Fig. 5c,d and Supplementary Table 5), which is the second-largest cluster of the VLPO$^{GABA}$ groups, expressed *Hcrtr2* RNA (Ox2R) (489 neurons; Supplementary Tables 6,7). *Hcrtr1* RNA (Ox1R) was not detected in any of the VLPO clusters, as previously reported by in situ hybridization[25] (Fig. 5b).

We confirmed these bioinformatic results using RNA scope in situ hybridization. Our in situ analysis confirmed an absence of expression of *Hcrtr2* mRNA in VLPO neurons that expressed both *Slc32a1* (*Vgat*) and *Gal* mRNAs (i.e., VLPO$^{GABA/Gal}$). In addition, *Hcrtr2* mRNA was expressed in 28.5% of VLPO neurons that express *Slc32a1* but not *Gal* mRNAs (i.e., VLPO$^{GABA}$) (Fig. 5e; Supplementary Table 8). These results confirm our in vitro recording findings that only VLPO$^{GABA}$ neurons, or a subset thereof, are capable of responding directly to Ox and hence represent a likely 'gateway' for Ox signaling into the cellular VLPO network.

**Inputs from orexin neurons excite VLPO$^{GABA}$ neurons.** We next verified functional synaptic connectivity between Ox and VLPO neurons using in vitro CRACM recordings. We expressed ChR2 in Ox neurons by injecting *AAV-DIO-ChR2-eYFP* or *AAV-DIO-ChR2-mCherry* into the lateral hypothalamus of *Ox-IRES-Cre* mice. We then recorded from VLPO neurons in brain slices while photostimulating the Ox axon terminals (Fig. 6a). We confirmed selective expression of *YFP* or *mCherry* in Ox neurons by immunolabeling (Fig. 1d and Supplementary Table 1). Histological assessment of the AAV injections confirmed that expression of ChR2 was restricted to the Ox field (Fig. 6b). Photostimulation of the Ox → VLPO input produced opto-evoked excitatory postsynaptic currents (oEPSCs) in 24% of recorded VLPO neurons ($n = 63$). DNQX (20–200 μM; $n = 4$) abolished the oEPSCs, indicating release of glutamate and AMPA-mediated signaling in VLPO neurons (Fig. 6c–e). Neurons that responded to photostimulations with glutamate-mediated oEPSCs displayed no changes in holding currents nor in resting membrane potentials during or after 60 s-long photostimulation trains indicating no detectable release of Ox (Fig. 6f).

To identify the postsynaptic target neurons in VLPO of this Ox input, we tested the VLPO neurons that responded to photostimulation for their response to NA and/or for expression of *Gal* by scRT-sqPCR ($n = 7$). We found that 6 of the recorded neurons were excited by NA (Fig. 6g) whereas 1 was inhibited. All VLPO neurons that responded to photostimulation of the Ox → VLPO input lacked *Gal* mRNA (Fig. 6h). Altogether, these results demonstrate that Ox nerve terminals exclusively contact and activate VLPO$^{GABA}$ neurons (Fig. 7).

**Discussion**
Here we demonstrate the existence of a functional polysynaptic circuit between the Ox neurons and sleep-promoting VLPO$^{GABA/Gal}$ neurons. We further demonstrate that selective activation of the Ox → VLPO$^{GABA}$ → VLPO$^{GABA/Gal}$ circuit is potently wake-promoting in vivo. And our in vitro results demonstrate that Ox directly activates Hcrtr2-expressing VLPO$^{GABA}$ neurons, but not VLPO$^{GABA/Gal}$ neurons. These results identify a clearly delineated polysynaptic circuit by which Ox neurons can effectively "turn off" sleep-promoting VLPO$^{GABA/Gal}$ neurons to promote arousal in vivo. Our findings specifically support a circuit model in which the Ox input (and co-released glutamate) exerts its excitatory influence on VLPO$^{GABA}$ interneurons, including in the

case of Hcrtr2-expressing VLPO$^{GABA}$ interneurons, which in turn inhibit sleep-promoting VLPO$^{GABA/Gal}$ neurons to promote and stabilize wakefulness. In this verified circuit model, loss of Ox, as occurs in narcolepsy, would result in a reduction in excitatory tone on VLPO$^{GABA}$ neurons, thereby reducing feedforward inhibition of VLPO$^{GABA/Gal}$ neurons and biasing their sustained activity and, hence, the sleep state.

A reciprocal inhibitory relationship between the VLPO and components of the brain's arousal system, including hypothalamic Ox neurons[26,27], is thought to support both the consolidation of behavioral state and the ability to rapidly and completely transition between behavioral states, e.g., wake to sleep[28]. This circuit arrangement has been conceptualized as being analogous to an electronic flip-flop switch, which possesses the desirable properties of being both self-stabilizing and highly stable in their 'on' or 'off' states. Hence, under normal neurobiological conditions, this flip-flop circuit arrangement ensures strong state boundary control, i.e., unstable intermediate states are avoided. However, when this circuitry is damaged, as occurs in various parasomnias and sleep-wake disorders such as narcolepsy, the 'weakened' switch operates closer to the boundary zone, leading to a greater percentage of time spent in unstable transition or intermediate states. In the case of narcolepsy, these frequent and unwanted transitions into the boundary zone result in the manifestation of debilitating physical symptoms including fragmentation of sleep or wake, cataplexy, sleep paralysis, hallucinations, and sleep-onset rapid-eye-movement sleep events (SOREMs). A similar phenomenon occurs during aging, when cell loss in the VLPO leads to weakening of the "sleep side" of the switch, in turn resulting in sleep fragmentation and daytime napping, both of which are frequent complaints in the elderly[29,30].

During wakefulness, sleep-promoting VLPO$^{GABA/Gal}$ neurons are inhibited by afferent inputs from many sources[16], including the Ox neurons. Intermingled however with VLPO$^{GABA/Gal}$ neurons are additional GABAergic neurons (VLPO$^{GABA}$) that lack galanin and appear to be active during wakefulness[12,18,31–33]. In this regard, they are similar to the GABA neurons in the more caudal lateral hypothalamus that also promote wakefulness by inhibiting the VLPO$^{GABA/Gal}$ neurons[13]. Here, we show that Ox directly excites a subset of Hcrtr2-expressing VLPO$^{GABA}$ neurons, and in turn through their local collaterals, these neurons inhibit the sleep-promoting VLPO$^{GABA/Gal}$ cell population. Findings from the present study, therefore, support a circuit model in which local VLPO$^{GABA}$ neurons function as an interface between afferent inputs to the VLPO and sleep-promoting VLPO$^{GABA/Gal}$ neurons. In other words, Ox, and possibly other wake-promoting signals, via feedforward inhibition, inhibit VLPO$^{GABA/Gal}$ neurons to produce arousal, and sleep-promoting signals, in turn via disinhibition, activate VLPO$^{GABA/Gal}$ neurons to produce sleep. This intra-VLPO circuit mechanism for promoting state transitions could be shared by most, if not all, wake- and sleep-promoting afferences that modulate VLPO activity[32,34–37]. Our model further predicts that VLPO$^{GABA}$ neurons are wake-active and, possibly, wake-promoting. In fact, recordings in the VLPO have uncovered neurons that are active in wakefulness[8,33] and optogenetic activation of Gad2-expressing VLPO neurons [which likely activates both VLPO$^{GABA/Gal}$ and VLPO$^{GABA}$ neurons] produces arousal, suggesting that co-activation of VLPO$^{GABA}$ and VLPO$^{GABA/Gal}$ neurons can override the ability of VLPO$^{GABA/Gal}$ neurons to produce sleep[38].

A better understanding of the synaptic, cellular, and circuit bases by which Ox neurons stabilize and maintain wakefulness has important implications for treating patients with neurological disorders associated with arousal dysfunction. For instance,

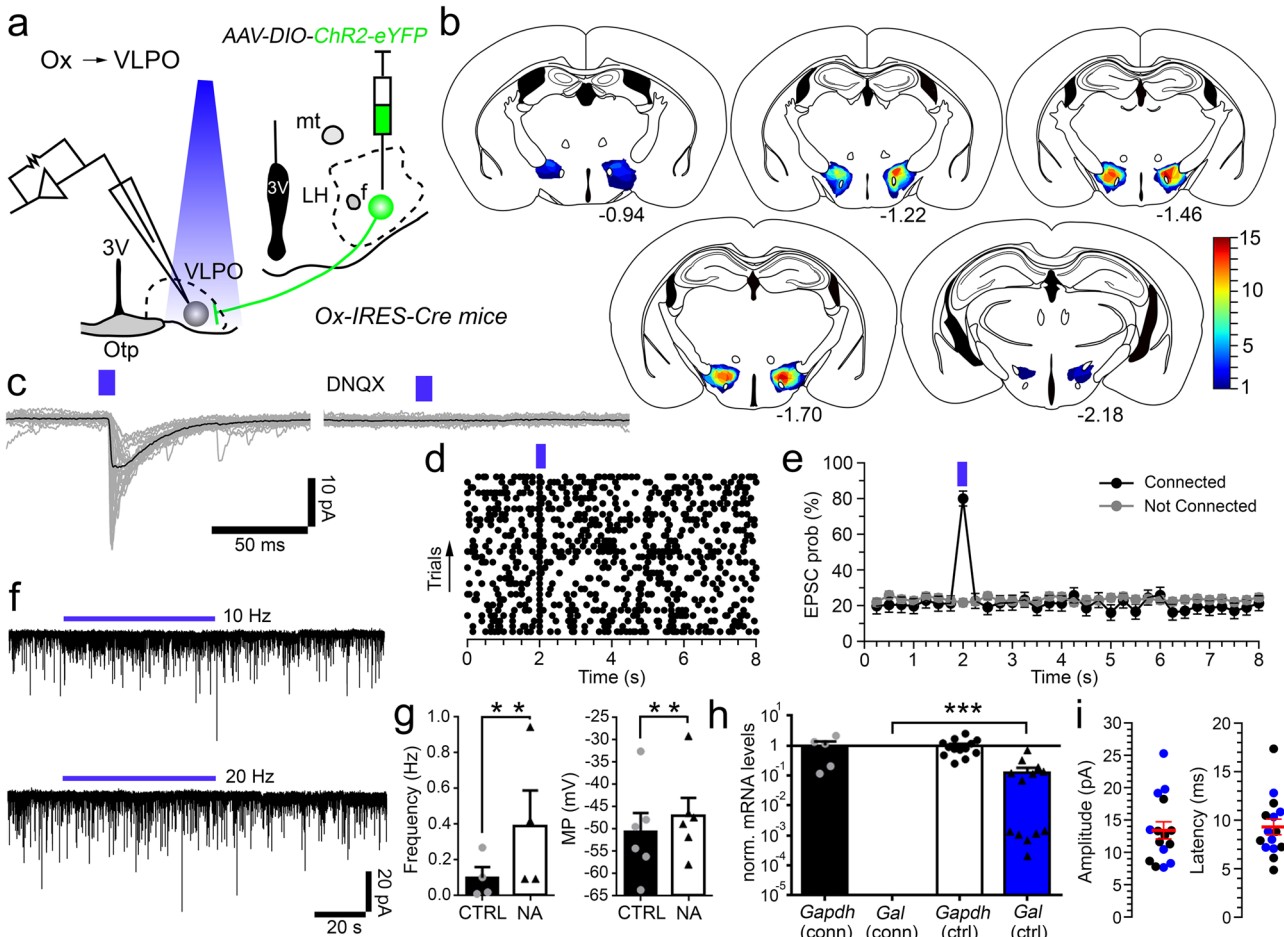

**Fig. 6 Orexin input excites VLPO^GABA neurons by glutamate release (Ox → VLPO^GABA). a** *AAV-DIO-ChR2-eYFP* injected into the Ox field of *Ox-IRES-Cre* mice and recordings from VLPO neurons. **b** Injection sites of 15 *Ox-IRES-Cre* mice used for in vitro CRACM ($n = 10$ mice) and in vivo optogenetics ($n = 5$ mice). **c** Photostimulation of Ox → VLPO input evokes AMPA-mediated oEPSCs in VLPO neurons (DNQX 200 µM; in grey: 30 individual oEPSCs; in black: average oEPSC). **d** Raster plot of EPSCs in a VLPO neuron before, during, and after photostimulation of the Ox → VLPO input (50 ms bins). **e** Average EPSC probability of VLPO neurons that responded ($n = 13$; black) and did not respond ($n = 48$; grey; means ± SEM) to the photostimulation of Ox input. **f** No changes in holding currents in response to photostimulation trains (60 s at 10 and 20 Hz; 10 ms light pulses) in neurons activated by the Ox → VLPO input with glutamate-mediated oEPSCs. The VLPO^GABA neurons are activated by the Ox → VLPO input. The VLPO neurons that responded to photostimulation of the Ox → VLPO input ($n = 7$) are excited by NA (**g**) and/or do not express *galanin* mRNA (**h**; scRT-sqPCR). Specifically 4 neurons were tested for both the responses to NA and for the presence of *Gal* mRNA, 2 only for NA, and 1 only for *Gal* mRNA. Mean effects of NA on the action potential frequency (**g** *left*; paired one-sided *t*-test, CTRL vs NA, $p = 0.0033$; $n = 4$) and membrane potential (g, *right*; paired one-sided *t*-test, CTRL vs NA, $p = 0.0017$; $n = 6$). Results from scRT-sqPCR for *Gapdh* and *galanin* (*Gal*) from 5 VLPO neurons Ox → VLPO input connected (conn) are compared to 13 controls (ctrl) VLPO^GABA/Gal neurons (**h**; ctrl: sampled from recorded VLPO^GABA/Gal TdTomato-labeled neurons from *Gal-IRES-Cre* mice injected with *AAV-DIO-TdTomato*). Mann-Whitney unpaired one-sided *t*-test, *Gal* (conn) vs *Gal* (ctrl); $p = 0.0001$. Values are normalized to the mean *Gapdh*. Means ± SEM. **i** oEPSC amplitude (*left*) and latency (*right*) in 15 VLPO neurons (blue dots: identified VLPO^GABA neurons; in red: means ± SEM). Recordings at $V_h = -70$ mV; 10 ms blue-light pulses (blue bars). ***$p < 0.001$ and **$p < 0.01$. 3V 3rd ventricle, Opt optical chiasm, f fornix, mt mammillothalamic tract, LH lateral hypothalamus. Atlas levels are per Franklin and Paxinos, 2001. For panels **e** and **g**: *n* refers to the number of recorded neurons. Panel d–e and g–i: Source Data file.

patients suffering from narcolepsy exhibit debilitating sleepiness, sleep attacks, and wake-instability, which take a tremendous toll on quality of life. At present, these symptoms are largely addressed clinically through the provision of stimulants, which themselves possess unwanted side effects. We show here that Ox exerts its wake-promoting effects, at least in part, through indirect inhibition of sleep-promoting VLPO^GABA/Gal neurons, and our electrophysiological and single-cell transcriptomics findings uncover and characterize two molecularly-distinct VLPO cell populations. There exists considerable redundancy in the brain's arousal circuitry, with only a few 'nodes' having been established as truly necessary for arousal maintenance[39–42], i.e., loss or

disruption produces a chronic reduction in arousal level. Similarly, Ox neurons likely produce arousal (and contribute to state stabilization) through multiple projections[14], wherein each post-synaptic 'target' of Ox neurons likely contributes to arousal control and stabilization, yet any given individual 'target' in isolation is unlikely necessary for arousal maintenance. Taken together, our findings suggest that VLPO^GABA neurons, or a subset thereof, may represent a 'common point of entry' for a wide range of inputs into the intra-VLPO cellular network. These findings also suggest the possibility of new 'targets' for the development of more selective pharmacologic strategies for treating the crippling inability to maintain consolidated

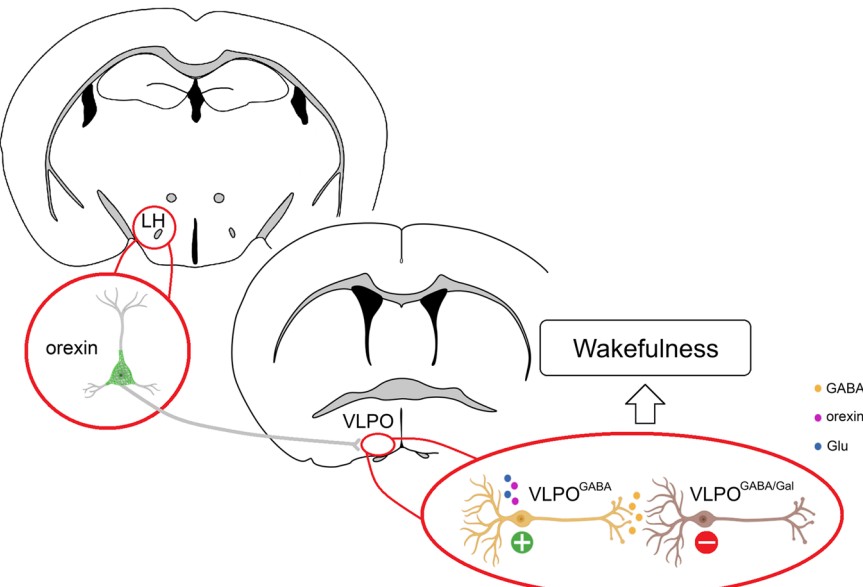

**Fig. 7 Orexin neurons promote wakefulness by inhibiting sleep-promoting VLPO$^{GABA/Gal}$ neurons via feed-forward inhibition.** Orexin neurons promote arousal through their projections to the VLPO. Orexin input to the VLPO directly activates VLPO GABAergic neurons that do not express galanin (VLPO$^{GABA}$) via co-release of Ox and glutamate. In turn, VLPO$^{GABA}$ neurons inhibit VLPO$^{GABA/Gal}$ sleep-promoting neurons to produce wakefulness.

wakefulness in narcolepsy and in other neurodegenerative, neurodevelopmental, and neuropsychiatric diseases in which disordered arousal, including even hyperarousal (e.g., insomnia), is a prominent clinical feature.

## Methods

**Animals and stereotaxic viral injections**. Mice were treated in accordance with guidelines from the National Institute of Health *Guide for the Care and Use of Laboratory Animals*. All protocols were approved by Beth Israel Deaconess Medical Center Institutional Animal Care and Use Committee and all efforts were directed to minimize the number of animals and their suffering. Mice were held at temperature of $22 \pm 1$ °C (40–60% of humidity) and on a 12 h light/dark cycle with available food and water access ad libitum.

We used adult mice, 6-12 weeks of age, and an equal proportion of male and female mice. We used: 55 C57BL/6J (WT); 33 *Vgat-IRES-Cre*; 25 *Galanin-IRES-Cre* (*Gal-IRES-Cre*); 11 *Orexin-IRES-Cre* (*Ox-IRES-Cre*) and 10 *Vgat-Flp::Gal-IRES-Cre* mice. WT mice were purchased from The Jackson Laboratory (Jax, Bar Harbor, ME). *Vgat-IRES-Cre* mice (obtained by The Jackson Laboratory: Jax Slc32a1$^{tm2(cre)}$ $^{Lowl}$/J, Cat. #016962) express Cre-recombinase under the control of vesicular GABA transporter (*Vgat, Slc32a1*) gene, to restrict the recombination in GABAergic neurons[43]. The *Gal-IRES-Cre* mice express Cre-recombinase under the *galanin* (*Gal*) gene promoter[9,44] and were given to us by Dr. Ramalingam Vetrivelan (Neurology Department, BIDMC, Boston, MA). The *Vgat-Flp::Gal-IRES-Cre* mice express *Flp*-recombinase in Vgat expressing neurons and Cre-recombinase in galanin expressing neurons and were obtained by crossing the *Slc32a1$^{tm1.1(flpo)Hze}$* mice (*Vgat-Flp*; obtained by The Jackson Laboratory: Jax B6.Cg-Slc32a1$^{tm1.1(flpo)Hze}$/J, Cat. #029591) with the *Gal-IRES-Cre* mice.

To activate the input from Ox neurons to VLPO (in vivo and in vitro recordings), we expressed ChR2-mCherry or ChR2-eYFP in the Ox neurons using two Cre-dependent AAVs: the *AAV-DIO-ChR2-mCherry (rAAV8-Ef1a-DIO-hChR2(H134R)-mCherry*; $1.5 \times 10^{13}$ virus molecules/ml; UNC Gene Therapy Center) and the *AAV-DIO-ChR2-eYFP (AAV10-Ef1a-DIO-hChR2(H134R)-eYFP*; $6.6 \times 10^{12}$ virus molecules/ml; provided by Dr. P.M.F.). As previously described[45] we placed two bilateral injections of *AAV-DIO-ChR2-mCherry* or *AAV-DIO-ChR2-eYFP* into the lateral hypothalamus of *Ox-IRES-Cre* mice (300 nl for each of the 4 injections, 2 per side: AP: −0.7 mm, DV: −5.00 mm, ML: ± 1.0 mm and AP: −1.6 mm, DV: −5.0 mm, ML: ± 0.8 mm). Animals used for the sleep studies were also instrumented for optogenetic stimulations and EEG/EMG recordings. EEG/EMG recordings started four weeks after the surgeries. Four to six weeks after the AAV injections we recorded in brain slices for in vitro CRACM recordings.

To identify VLPO GABAergic and galanin neurons in recording slices we labeled them with TdTomato or GFP by placing bilateral microinjections of *AAV-DIO-TdTomato (AAV10-Ef1a-DIO-TdTomato*; was provided by Dr. P.M.F.) or *AAV-DIO-GFP (rAAV8- DIO-GFP*; $6 \times 10^{12}$ virus molecules/ml; prepared by the UNC Gene Therapy Center, University of North Carolina, Chapel Hill, NC) into the VLPO (180 nl; AP: 0.26 mm, DV: −5.2 mm, ML: ± 0.6 mm) of *Vgat-IRES-Cre, Gal-IRES-Cre* and *Vgat-Flp::Gal-IRES-Cre* mice. Two weeks after the AAV

injections, we recorded in brain slices from fluorescently labeled Vgat or galanin expressing neurons.

To study the VLPO internal circuit by in vitro CRACM recordings, we used two Cre-dependent AAVs: one AAV that codes for ChR2 (*AAV-DIO-ChR2-mCherry*) and one that codes for TdTomato (*AAV-DIO-TdTomato*). We also used a Flp-dependent AAV that codes for ChR2-eYFP, the *AAV-fDIO-ChR2-eYFP (AAVDJ-Ef1a-fDIO-ChR2-eYFP*; $2 \times 10^{12}$ virus molecules/ml; UNC Gene Therapy Center). Specifically, to activate collateral inputs between VLPO galanin neurons, we unilaterally injected *AAV-DIO-ChR2-mCherry* (35–50 nl) into the VLPO of *Gal-IRES-Cre* mice (AP: 0.26 mm, DV: −5.2 mm, ML: ± 0.6 mm) and we recorded from VLPO galanin neurons. To activate the input from local VLPO GABAergic neurons to VLPO galanin neurons, we unilaterally injected the Flp-dependent *AAV-fDIO-ChR2-eYFP* (35-50 nl) and the Cre-dependent *AAV-DIO-TdTomato* (180 nl) into the VLPO of *Vgat-Flp::Gal-IRES-Cre* mice (AP: + 0.26 mm, DV: −5.2 mm, ML: ± 0.6 mm). This results in the expression of ChR2-eYFP in the GABAergic neurons and TdTomato in galanin neurons. We recorded from VLPO galanin neurons labeled by TdTomato. Stereotaxic coordinates for all the AAV injections were based on mouse brain atlas by Paxinos and Franklin (2001). We conducted slice recordings 4-6 weeks after the AAV injections.

**Generation of *Ox-IRES-Cre* knock-in mice**. These mice were generated, validated, and kindly provided by Drs. D.K., T.M., B.B.L., and T.E.S. The *Ox-IRES-Cre* mice express the Cre-recombinase under the control of the endogenous orexin/hypocretin (*Ox/Hcrt*) gene locus. Briefly, the generation of the gene targeting construct was performed through the help of a modified BAC clone[46] that contained the *Ox/Hcrt* genomic sequence, as previously reported[47]. This BAC-derived construct spans the region 5 kb upstream of the *IRES* site and 2 kb downstream of the *Cre* site inserted into 3′-end genomic region. Then, it was electroporated into mouse embryonic stem (ES) cells (W4/129S6, Taconic, Germantown, NY). Correctly targeted ES cells were identified and then injected into blastocysts to generate chimeras. The male chimeras mice were bred to mice bearing a Flp-recombinase transgene, in order to remove the selection marker neomycin. The Cre-recombinase activity was verified in the hypothalamus by crossing male chimeras to Ai14 Cre reporter line (obtained by The Jackson Laboratory: Jax B6;129S6-Gt(ROSA)26Sor$^{tm14(CAG-tdTomato)Hze}$/J, Cat. #007908)[47].

**EEG/EMG and optical fiber implants**. We used *Ox-IRES-Cre* ($n = 5$ mice) and WT ($n = 3$ mice as controls). Following bilateral injections of *AAV-DIO-ChR2-mCherry* into the lateral hypothalamus mice underwent a second surgery to implant a headstage for EEG and EMG recordings and optical fibers for bilateral optogenetic stimulation of the Ox input to VLPO. The headstages were custom-made in-house by assembling a 6-pin connector (Heilind Electronics, catalog #MMX853-43-006-10-001000), 4 EEG screws (Pinnacle, catalog# 8403), and 2 flexible EMG wire electrodes (Plastics One, catalog #E363). During surgery, two burr holes were drilled in the skull of the anesthetized mice immediately above the VLPO (AP = + 0.4 mm; ML = ± 0.7 mm) for placement of the optical fibers. Each optical fiber was stereotaxically guided into position (DV = −5.0 mm). We drilled 4 additional burr holes (0.7 mm diameter) for EEG electrode placement. The EMG

electrodes were then guided down the back of the neck underneath the trapezius muscle. EEG and EMG electrodes and optical fibers were glued into place using a mixture of dental cement and cyanoacrylate glue to provide insulation and structural stability. The optical fibers were manufactured in-house by assembling a fiber optic cable (200 μm outer diameter, ThorLabs, Cat. #FT200EMT) inserted into a ferrule (ceramic, 200 μm internal diameter, ThorLabs, Cat. CFLC230-10) and epoxied into place (Precision Fiber Products, Chula Vista, CA, Cat. #353ND). The optical fiber was cut to size (5.5 mm) and then flat cleaved and polished at the mating end of the ferrule.

**Sleep-wake monitoring, signal processing, optogenetic experiments and immunohistochemistry.** Sleep-wake recordings were conducted 2 weeks after surgeries. For recordings the mice were housed individually in transparent barrels in an insulated sound-proofed recording chamber maintained at an ambient temperature of $22 \pm 1\,°C$ and on a 12 h light/dark cycle (lights-on at 6 AM, Zeitgeber time: ZT0) with food and water available *ad libitum*. Mice were habituated for at least 3 days before commencing polysomnographic recordings. Cortical EEG (ipsilateral frontoparietal leads) and EMG signals were amplified x5000 (Model 3600; AM Systems, Sequim, WA) and digitized with a resolution of 500 Hz using a Micro 1401-3 (Cambridge Electronic Design; Cambridge, UK).

Offline, EEG signals were digitally filtered using a $2^{nd}$ order Butterworth, band-pass, zero-phase filter with cut-off frequencies 0.5 Hz and 225 Hz filtfilt function, MATLAB (version R2020B; Mathworks, Natick, MA). EMG signals were likewise filtered using cut-off frequencies 100 Hz and 225 Hz. Also, for the trial-averaged analysis of EMG responses to light stimulation, EMG time series were integrated, smoothed (MATLAB; smooth function; 500-sample window), and normalized according to the pre-stimulus mean. Continuous wavelet transform of EEG signals were performed using MATLAB based tools developed by Grinsted and colleges (Available for download at: https://github.com/grinsted/wavelet-coherence).

For optogenetic experiments, experiments were carried out between ZT3-ZT9 (a period of low activity in nocturnal mice). We modified an open-source Online Sleep Detection script (Spike 2, Cambridge Electronic Design, source code available from the Cambridge Electronic Design website) to detect when mice had been in NREM sleep or REM sleep for at least 20 s, to trigger a 12-15 mW, 473 nm blue light (R471003GX, LaserGlow, Toronto, Canada) stimulation to the VLPO. The stimulation consisted of light pulses (5 ms) delivered at frequencies 1, 5, 10, 20 Hz over a 10 s period, delivered in random order throughout the recording session. A refractory period of at least 3 min was allowed between each stimulation, no mouse received more than 200 stimulations over any recording session and each recording session was separated by at least two days.

Following the recordings, we intracardially perfused the mice with formalin (10% buffered solution) under deep anesthesia with isoflurane (5% in $O_2$). Brains were post-fixed in formalin (10% buffered solution, overnight), cryoprotected (20% sucrose), and cut into 40 μm sections. To verify the location of *AAV-DIO-ChR2-mCherry* injections sites, we incubated the sections in rabbit anti-ds Red primary antibodies (1:3000: overnight; Cat. #: 632496; Lot #: 1904182; Takara Bio USA, Mountain View, CA) and then in Alexa Fluor-555-conjugated donkey anti-rabbit secondary antibodies (1:500; Cat. #: A31572; Lot #: 2286312; 2hr; Invitrogen). We mapped the distribution of the ChR2- mCherry expressing neurons for each recorded animal as described in the methods above. No ChR2-mCherry expression was found in WT mice ($n = 3$).

**Sleep scoring, analysis and statistics.** We used a custom-written MATLAB script for scoring arousals within stimulation periods on a trial-by-trial basis. The scorer was presented with a 30 s epoch of EEG and EMG recording centered on the stimulation period. The scorer was blinded to the animal genotype and stimulation condition (i.e., sham versus 1, 5, 10, or 20 Hz). The scorer determined, through visual inspection, if arousal occurred within the 10s-long stimulation period. Scored arousals required an EEG change lasting at least three seconds. We excluded trials where the 10 s pre-stimulus period contained a mixture of states (e.g., micro-arousals, muscle twitching activity, NREM-to-REM transitory states). Sham stimulation trials were interleaved amongst the real laser stimulations.

To examine the time course of evoked EEG and EMG responses to light stimulation, we computed the continuous wavelet transform of the EEG and the integral of the EMG for each stimulation trial, inclusive of the 10 s before and the 20 s after the stimulation period. Mean EEG wavelet power and integrated EMG signals from each animal were subsequently averaged ($n = 5$). For the mean EMG signal we calculated the bootstrap confidence interval (MATLAB; *bootci* function). For average EEG power plots, confidence interval contours were computed as follows. For each animal we calculated the difference in wavelet power (real minus sham) across the time-frequency plane (($-10$ to $+30$ s) x (0.5–30 Hz)). The data was smoothed in the time-domain (500-sample-wide moving average) and were subsequently down-sampled by a factor of 10. We used bootstrap resampling, with replacement, of the paired difference plots (all 3,125 permutations) to calculate the 95% bootstrap confidence interval of the mean power difference at each time-frequency coordinate. 5% significance contours were drawn around regions in the spectral plane where effect size confidence intervals excluded zero. Very small clusters of significance were removed for clarity and statistical conservatism.

All parametric statistical testing were carried out using Sigmastat (version 4.0, Systat Software Inc. San Jose, CA). Paired bootstrap resampling was performed using MATLAB; in each case, 5000 bootstrap samples were taken, and confidence intervals were bias-corrected and accelerated. The details of the statistical tests used for each experiment may be found in the results. Effect sizes were considered statistically significant for $p < 0.05$. All experimental data were subject to histological validation. Data were excluded if the conditions of the histological validation were not met (i.e., cases in which there was not adequate bilateral transduction of the viral vector or optical fiber placement was not correctly positioned). All behavioral recordings were scored by an investigator that was blinded to the recording conditions. We represented data as mean ± SEM and $n$ refers to the number of animals per group.

**Brain slice preparations and in vitro electrophysiological recordings.** We deeply anesthetized mice with isoflurane via inhalation (5% in O2) and transcardially perfused them with ice-cold modified ACSF (N-Methyl-D-glucamin-(NMDG)-based ACSF solution). We quickly removed the mouse brains and sectioned them in coronal slices (250 μm-thickness) in ice-cold NMDG-based ACSF using a vibrating microtome (VT1200S, Leica, Bannockburn, IL). We incubated the recording slices first for 5 min at 37 °C in NMDG-based ACSF, then in normal ACSF (Na-based solution) for an additional 10 min at 37 °C, and then we let them return to room temperature (RT) for at least 1 h.

We recorded VLPO neurons in brain slices submerged and perfused (1–1.5 ml/min) with Na-based ACSF. We recorded under infrared differential interference contrast (IR-DIC) visualization and we recorded fluorescently labeled neurons using a combination of fluorescence and IR-DIC microscopy. We used a fixed-stage upright microscope (BX51WI, Olympus America Inc.) equipped with a Nomarski water immersion lens (Olympus 40X/0.8NAW) and an IR-sensitive CCD camera (ORCA-ER, Hamamatsu, Bridgewater, NJ). Real-time images were acquired using MATLAB (MathWorks) script software. We recorded VLPO neurons in whole-cell and cell-attached configurations using a Multiclamp 700B amplifier (Molecular Devices, Foster City, CA), a Digidata 1322 A interface, and Clampex 9.0 software (Molecular Devices). Neurons showing changes in input resistance of more than 10% over time, were excluded from the analysis. We recorded action potential firing in Na-based ACSF in cell-attached (voltage-clamp mode; Vh = 0 mV) or in whole-cell (current-clamp mode) configurations using the K-gluconate-based solution. For cells that were not spontaneously active we raised the ACSF KCl concentration from 2.5 to 6.3 mM.

For the CRACM experiments we photostimulate ChR2 expressing neurons, axons, and terminals in VLPO using full-field light pulses (~10 mW/mm², 1 mm beam width) from a 5 W Luxeon blue light-emitting diode (470 nm wavelength; Thorlabs, Newton, NJ; Cat. #M470L2-C4) coupled to epifluorescence pathway of microscope. We photostimulated with 10 ms light pulses (0.1 Hz, for a minimum of 30 trials). In some recordings, we used 60 s trains at 10 and 20 Hz (10 ms light pulses).

We recorded spontaneous inhibitory postsynaptic currents (sIPSCs) and opto-evoked IPSCs (oIPSCs) in Na-based ACSF using a Cs-methane-sulfonate-based pipette solution at Vh = 0 mV. We recorded $VLPO^{GABA/Gal}$ neurons while photostimulating the local $VLPO^{GABA/Gal} \rightarrow VLPO^{GABA/Gal}$ and $VLPO^{GABA} \rightarrow VLPO^{GABA/Gal}$ inputs at the reversal potential of the ChR2-mediated current[19,20] determined for each cell (Vh = $-5$ to $-15$ mV)· We recorded opto-evoked excitatory postsynaptic currents (oEPSCs) in Na-based ACSF ( $V_h = -70$ mV) using the K-gluconate-based pipette solution. In all the recordings we added 0.5% biocytin in the pipette solutions to mark the recorded neurons.

**Solutions and Chemicals.** The NMDG-based ACSF contained (in mM): 100 NMDG, 2.5 KCl, 1.24 $NaH_2PO_4$, 30 $NaHCO_3$, 25 glucose, 20 HEPES, 2 thiourea, 5 Na-L-ascorbate, 3 Na-pyruvate, 0.5 $CaCl_2$, 10 $MgSO_4$ (pH 7.3 with HCl with 95% $O_2$ and 5% $CO_2$; 310–320 mOsm). The Na-based-ACSF contained (in mM): 120 NaCl, 2.5 KCl, 1.3 $MgCl_2$, 10 glucose, 26 $NaHCO_3$, 1.24 $NaH_2PO_4$, 4 $CaCl_2$, 2 thiourea, 1 Na-L-ascorbate, 3 Na-pyruvate (pH 7.3–7.4 with 95% $O_2$ and 5% $CO_2$; 310–320 mOsm). The K-gluconate-based solution contained (in mM): 120 K-Gluconate, 10 KCl, 3 $MgCl_2$, 10 HEPES, 2.5 K-ATP, 0.5 Na-GTP (pH 7.2 adjusted with KOH; 280 mOsm). The Cs-methane-sulfonate-based pipette solution contained (in mM): 125 Cs-methane-sulfonate, 11 KCl, 10 HEPES, 0.1 $CaCl_2$, 1 EGTA, 5 Mg-ATP and 0.3 Na-GTP (pH 7.2 adjusted with CsOH, 280 mOsm).

We purchased Ox-A from Bachem (Bubendorf, Switzerland), noradrenalin, tetrodotoxin, and kynurenic acid from Cayman Chemical (Ann Arbor, MI), carbachol, bicuculline methiodide from Tocris Bioscience (Ellisville, MO). We purchased all other chemicals from Fisher Scientific (Waltham, MA) or Sigma-Aldrich (Saint Luis, MO).

**Data analysis and statistics for in vitro recordings.** We analyzed recording data using Clampfit 10 (Molecular Devices), MiniAnalysis 6 (Synaptosoft, Leonia, NJ), Python 3 (www.python.org) and MATLAB (version R2020B; MathWorks) software. We prepared the figures using Igor Pro 6 (WaveMetrics), Prism 7 (GraphPad, La Jolla, CA), Inkscape (GitLab), Photoshop (Adobe), and BioRender software.

To ensure unbiased detection of the synaptic events, IPSCs and EPSCs were detected and analyzed automatically using MiniAnalysis. We measured synaptic event frequency and amplitude in control ACSF (last 5 min before drug applications), drug application (last 5 min of 10–15 min of drug applications), and in washout (last 5 min of 30–40 min washouts).

We used the nonparametric Kolmogorov-Smirnov test (K-S test; MiniAnalysis) to evaluate the responses of drugs on sIPSC frequency or amplitude (statistical significance for $p < 0.05$, K-S test). We statistically compared mean sIPSC inter-event interval cumulative distributions using two-way repeated-measures (RM) ANOVA followed by Bonferroni's multiple comparisons *post-hoc* test. We considered VLPO neurons to be responsive to photostimulation if the oEPSC or oIPSC probability during the first 50 ms that follows the light pulses was greater than baseline EPSC/IPSC probability + five times the SEM (baseline EPSC probability = $22.43 \pm 1.84\%$, $n = 61$; baseline IPSC probability = $9.08 \pm 1.83$, $n = 14$)[23]. We calculated the latency of the oEPSCs and oIPSCs as the time difference between the start of the light pulse and the 5% rise point of the first synaptic event[48].

We calculated firing frequency and membrane potential changes by comparing values in control ACSF (last 2 min before drug applications), drug application (last 2 min of the 4–6 min of drug applications), and in washout (last 2 min of the 15–20 min washout).

We represented data as mean ± SEM and $n$ refers to the number of cells per group, unless otherwise specified. We compared group means using one-way or two-way RM ANOVA followed by Bonferroni's multiple comparisons *post-hoc* test (*adjusted-p-value, adj-p*) or paired *t*-tests. Values showing $p < 0.05$ were considered significant.

**Peak-scaled non-stationary fluctuation analysis**. We used peak-scaled non-stationary fluctuation analysis to determine changes in GABA$_A$ single-channel current ($i$) or in the number of GABA$_A$ channels activated ($N$) in response to Ox-A[21–23,49]. As previously described[23], we low-pass filtered and then aligned the peaks of photo-evoked IPSCs. We scaled the averaged mean-current waveform to the peak amplitude of individual oIPSCs, squared the difference, and then sampled this variance time series in 30 bins of equal current decrement from peak to baseline. The binned variance, was plotted against the mean-current amplitude (Fig. 4j). We estimated the $N$ and the $i$ values by least-squares fitting of the peak-scaled variance and mean-current curve to the equation: $\delta^2 = iI - I^2/N + b$ where $\delta^2$ is the variance, $I$ is the mean-current, and $b$ is baseline variance. For each cell, we selected a minimum of 20 oIPSCs that had no overlapping spontaneous IPSCs. All the analysis was done using software written in Python 3.

**Immunohistochemistry, in situ hybridization and RNA scope in situ hybridization**. Following recordings, we fixed the recorded and adjacent slices, and the block of brain containing the AAV injection sites in formalin (10% buffered solution, overnight). The blocks containing the injection sites were then cryoprotected (20% sucrose) and recut in 40 μm sections with a sliding microtome (Microm HM 440E, GMI Inc., Minneapolis, MN). We processed the recorded slices with streptavidin-conjugated Alexa Fluor-488 (green; 1:500; Cat# S32354 and Lot# 1719656) or Alexa Fluor-555 (orange-red; 1:500; Cat# S21381 and Lot# 1010095; overnight; Invitrogen, Thermo Fisher Scientific Waltham, MA; Fig. 2, Supplementary Figs. 4 and 6). We imaged the recorded fluorescently labeled neurons using a confocal microscope (Zeiss, LSM 5 Pascal) and Zen 2009 software (Zeiss) and we mapped their distribution onto template drawings[50].

We mapped the distribution of the ChR2-mCherry and ChR2-eYFP expressing neurons for each recorded animal by enhancing mCherry and YFP fluorescence by immunolabeling. For ChR2-mCherry immunolabeling we used rabbit anti-ds Red primary antibodies (1:3000; overnight; Cat. #: 632496; Lot #: 1904182; Takara Bio USA) and Alexa Fluor-555-conjugated donkey anti-rabbit secondary antibodies (1:500; Cat. #: A31572; Lot #: 2286312; overnight; Invitrogen). For ChR2-eYFP immunolabeling we used chicken anti-GFP primary antibodies (1:2000; overnight; Cat. #: A10262; Lot #: 2156242; Invitrogen) and Alexa Fluor-488-conjugated goat anti-chicken secondary antibodies (1:500; Cat. #: A11039; Lot #: 2304258; overnight; Invitrogen). We wet mounted and we scanned the whole sections using a slide scanner (Olympus VS120) or a confocal microscope (Zeiss LSM 5 Pascal) at a final magnification of 20X. We used a z-stack at 3–8 μm intervals to image throughout the slice. We viewed stacks of images using OlyVIA (Olympus) or Zen 2009 (Zeiss) software to identify the region containing transduced somata. We plotted the outline regions containing ChR2-mCherry or ChR2-eYFP labelled neurons onto template drawings[50] using Photoshop (Adobe). We compiled the distribution of ChR2-mCherry or ChR2-eYFP expressing neurons in a color map using a Python script[51], representing the region of maximum number of overlapping cases in deep red, with the hues to green, light blue, to deep blue indicating fewer (Fig. 4 and Fig. 6).

We validated the *AAV-DIO-ChR2-mCherry* and the *AAV-DIO-ChR2-eYFP* in *Ox-IRES-Cre* mice by immunolabeling. We processed for Ox immunoreactivity slices from *Ox-IRES-Cre* mice that were injected in the lateral hypothalamus with *AAV-DIO-ChR2-mCherry* ($n = 2$ mice) and *AAV-DIO-ChR2-eYFP* ($n = 2$ mice; Fig. 1 and Supplementary Table 1). Sections (40 μm) were only processed for immunohistochemistry for Ox whereas mCherry native fluorescence was used for

the ChR2-mCherry labelling. We incubated the sections in goat anti-Ox-A (1:500; Cat. #: sc-8070; Lot #: A2915; Santa Cruz Biotechnology, Dallas, TX) primary antibodies (overnight) and then in Alexa Fluor-488-conjugated donkey anti-goat (1:500, Cat. #: A11055; Lot #: 1942238; Invitrogen) secondary antibodies (2 h)[23]. Sections from mice injected with *AAV-DIO-ChR2-eYFP* were processed for immunohistochemistry for both Ox and YFP. We incubated the sections in goat anti-Ox-A (1:500; Cat. #: sc-8070; Lot #: A2915; Santa Cruz Biotechnology) and chicken anti-GFP (1:2000; Cat. #: A10262; Lot #: 2156242; Invitrogen) primary antibodies (overnight) and then in Alexa Fluor-555-conjugated donkey anti-goat (1:500; Cat. #: A21432; Lot #: 1697092; Invitrogen) and Alexa Fluor-488-conjugated rabbit anti-chicken (1:500; Cat. #: NB710-94968; Lot #: 143-090; Novus Biologicals, Littleton, CO) secondary antibodies (2 hr). We counted the ChR2-mCherry expressing neurons (native fluorescence) and the ChR2-eYFP expressing neurons (immunolabeled in green with Alexa-Fluor-488) that were double-labeled for Ox immunoreactivity using a confocal microscope (Zeiss LSM 5 Pascal) and Zen 2009 software (Zeiss).

To visualize the Ox innervation of VLPO we processed for Ox immunolabeling sections from WT mice ($n = 4$ mice). We intracardially perfused the mice with formalin (10% buffered solution) under deep anesthesia with isoflurane (5% in $O_2$). Brains are post-fixed in formalin (10% buffered solution, overnight), cryoprotected (20% sucrose), and cut into 40 μm sections into 3 series using a sliding microtome (Microm HM 440E, GMI Inc., Minneapolis, MN). We incubated the sections in goat anti-Ox-A (1:500; Cat. #: sc-8070; Lot #: A2915; Santa Cruz Biotechnology) primary antibodies (overnight) and then in Alexa Fluor-555-conjugated donkey anti-goat (1:500; Cat. #: A21432; Lot #: 1697092; Invitrogen) secondary antibodies (2 h)[23]. We imaged and photographed the covered sections (Vectashield mounting medium; Vector Laboratories, San Diego, CA) visualized and photographed the sections using a confocal microscope (Zeiss LSM 5 Pascal), and Zen 2009 software (Zeiss).

We performed DIG-labeled RNA probe in situ hybridization for *Vgat* (*Slc32a1*) mRNA in *Vgat-IRES-Cre* mice injected into the VLPO with *AAV-DIO-TdTomato* ($n = 2$ mice) or with *AAV-DIO-GFP* ($n = 2$ mice; Supplementary Fig. 5 and Supplementary Table 1)[23]. Four weeks after AAV injections, we intracardially perfused the mice with formalin (10% buffered solution) under deep anesthesia with isoflurane (5% in $O_2$). Brains were post-fixed in formalin (10% buffered solution, overnight), cryoprotected (20% sucrose), and cut into 40 μm sections into 3 series using a sliding microtome (Microm HM 440E, GMI Inc). Sections were washed in RNAse-free PBS, containing diethylpyrocarbonate (DEPC) and then incubated in hybridization buffer at 53 ˚C (1 h). The Vgat probe was denatured at 80˚C (10 min) and then added to the hybridization buffer and incubated overnight at 53 ˚C. The sections were washed in saline citrate (SSC) with 50% formamide at 53 ˚C (2 h), washed in tris buffered saline (TBS; pH 7.5; 30 min), incubated in 1% blocking reagent (Roche Applied Science, Penzberg, Germany; 30 min), incubated in peroxidase-conjugated DIG antibodies (1:500, Cat. #: 11207733910; Lot #: 13296300; Roche Applied Science, now Millipore Sigma; overnight) and then washed in TBS. Sections from mice injected with *AAV-DIO-GFP* were labeled with Cy3 (red) by Tyramide signal amplification (1:50, Perkin Elmer, Waltham, MA; 30 min) then immunolabeled for GFP with chicken anti-GFP primary antibodies (1:2000; Cat. #: A10262; Lot #: 2156242; Invitrogen; overnight) and then in Alexa Fluor-488-conjugated goat anti-chicken secondary antibodies (1:500; Cat. #: A11039; Lot #: 2304258; Invitrogen; 2 h). Sections from mice injected with *AAV-DIO-TdTomato* were labeled with Cy5 (far-red) by Tyramide signal amplification (1:50, Perkin Elmer; 30 min) then immunolabeled for TdTomato with rabbit anti-ds Red primary antibodies (1:500, Cat. #: 632496; Lot #: 1904182; Takara Bio USA; overnight) and then in Alexa Fluor-555-conjugated donkey anti-rabbit secondary antibodies (1:500; Cat. #: A31572; Lot #: 2286312; Invitrogen; 2 hr). Sections were washed first in TBS and then in PBS, mounted, and coverslipped in deionized water. We imaged and photographed the sections using a confocal microscope (Zeiss LSM 5 Pascal). The Vgat probe was provided by Dr. Shigefumi Yokota (University School of Medicine, Izumo, Japan)[23].

For RNA scope in situ hybridization for *Gal*, *Vgat* and *Hcrtr2* mRNAs, we used RNA Scope Multiplex Fluorescent Reagent Kit V2 Advanced Cell Diagnostics, Hayward, CA; Cat. #323100. We intracardially perfused the mice with formalin (10% buffered solution) under deep anesthesia with isoflurane (5% in $O_2$). Brains were post-fixed in formalin (10% buffered solution, overnight), cryoprotected (20% sucrose), and cut into 30 μm sections. The sections were treated with hydrogen peroxide (20 min; RT). We then performed a target retrieval procedure at 100 ˚C (5 min), treated the sections with protease (40 ˚C; 30 min; Protease III, room temperature) and incubated them with RNA scope probes for *galanin*-C1 (RNA scope® Probe- Mm-*Gal*; Cat. #400961), *Vgat*-C2 (RNA scope® Probe- Mm-*Slc32a1*; Cat. #319199) and *Hcrtr2*-C3 (RNA scope® Probe- Mm-*Hcrtr2-O1*; Advanced Cell Diagnostics; Cat. #581631) for the hybridization step (2 hr; 40 ˚C). After hybridization, we performed the amplification steps (40 ˚C; AMP1-FL and AMP2-FL: 30 min each; AMP3-FL: 15 min), followed by horse radish peroxidase-C1 (HRP-C1) amplification (40 ˚C for 15 min). Sections were then incubated in TSA plus Cy3 (Perkin Elmer; Cat. #NEL744001KT) to visualize *Gal* (Channel 1 at 550 nm) mRNA in red. This is followed by incubating the sections in HRP-C2 amplification step (40 ˚C; 15 min). Sections were then incubated in TSA plus Fluorescein (Perkin Elmer, Cat. #: NEL754001KT) fluorophore (1:1000; 30 min) to visualize *Vgat* (Channel 2 at 488 nm) in green. In the last step of the process, sections were

subjected to HRP-C3 amplification (40 °C; 15 min) followed by TSA plus Cy5 incubation (40 °C; 30 min; Perkin Elmer; Cat. #NEL754001KT) to visualize Hcrtr2 (Channel 3 at 647 nm) mRNA in magenta. After each fluorophore step, sections were subjected to HRP blocking (40 °C; 15 min). After each step in the protocol, we washed the sections two times with 1X wash buffer provided in the kit.

We combined RNA scope in situ hybridization for Vgat mRNA (in Vgat-Flp mice) and for Gal mRNA (in Gal-IRES-Cre mice) with immunolabeling for YFP and TdTomato (Supplementary Fig. 8 and Supplementary Table 1). Mice were injected into the VLPO with AAV-fDIO-ChR2-eYFP (n = 2 Vgat-Flp mice) or with AAV-DIO-TdTomato (n = 2 Gal-IRES-Cre mice). Four weeks after AAV injections, we intracardially perfused the mice with formalin (10% buffered solution) under deep anesthesia with isoflurane (5% in $O_2$). Brains were post-fixed in formalin (10% buffered solution, overnight), cryoprotected (20% sucrose) and cut into 30 µm, and processed for RNA scope in situ hybridization for Vgat mRNA following the protocol described above. For the sections from Vgat-Flp mice injected with AAV-fDIO-ChR2-eYFP the Vgat mRNA was labeled with Cy3 (in red) and then the sections were immunolabeled for GFP (in green) with chicken anti-GFP primary antibodies (1:2000; Cat. #: A10262; Lot #: 2156242; Invitrogen; over 2 nights) and then in Alexa Fluor-488-conjugated goat anti-chicken secondary antibodies (1:500; Cat. #: A11039; Lot #: 2304258; Invitrogen; 4 hr). For the sections from Gal-IRES-Cre mice injected with AAV-DIO-TdTomato the Gal mRNA was labeled with Fluorescein fluorophore (in green) and then the sections were immunolabeled for TdTomato with rabbit anti-ds Red primary antibodies (1:1000, Cat. #: 632496; Lot #: 1904182; Takara Bio USA; over 2 nights) and then in Alexa Fluor-555-conjugated donkey anti-rabbit secondary antibodies (1:500; Cat. #: A31572; Lot #: 2286312; Invitrogen; 4 hr).

We imaged and photographed the covered sections (Vectashield mounting medium; Vector Laboratories) with a confocal microscope (Leica Stellaris 5) at final magnification of 20X and 63X. We used a z-stack at 3–5 µm intervals to image throughout the slice after tile acquisition. We viewed stacks of images using image J software to identify the region containing the VLPO and the fluorescent signals for RNA scope and immunohistochemistry at cellular level.

**Single cell RT-PCR and RT-sqPCR**. To identify the phenotype of recorded neurons, we performed single-cell reverse transcription-polymerase chain reaction (scRT-PCR) and semi-quantitative scRT-PCR (scRT-sqPCR). After completing whole-cell recordings, we harvested the cytoplasm of the recorded VLPO neurons (n = 26) through aspiration controlled under fluorescent and IR-DIC visualizations. We then performed cDNA synthesis as previously described[52] using the single-cell-to-CT^TM kit (Ambion, Thermo Fisher Scientific). We first converted the mRNA into cDNA (9 µl of Single Cell Lysis buffer and 1 µl of DNAase I solution to each extracted sample stored in RNAse-free PCR tubes; room temperature; 5 min), stopped the reaction (1 µl of Stop solution; room temperature; 2 min) and stored the samples (−20 °C; up to two months). For the pre-amplification protocol, we added 3 µl VILO and 1.5 µl Cell Superscript to each sample (25 °C; 10 min; 42 °C; 60 min and 85 °C; 5 min). We then added (PreAmp Mix 5 µl and of Taqman Gene Expression Master mix 6 µl to each sample; 95 °C; 10 min). We repeated 14 cycles (95 °C; 15 s and 60 °C; 4 min). Samples were then processed for scRT-PCR or scRT-sqPCR.

For scRT-PCR, amplification of cDNA encoding for the galanin (Gal), glyceraldehyde 3-phosphate dehydrogenase (Gapdh), glutamate decarboxylase Gad2 (Gad65) and Gad1 (Gad67) were performed in two rounds amplifications (40 cycles each). For the second amplification we used a 1:6 dilution of the first amplification material. Gal 1st amplification primers: GalF: 5′-GGCTGGCTCCTGTTGGTTGT-3′ and GalR: 5′- GCTCTCAGGCAGGGGCAC-3′ (annealing temperature; AT: 57.2 °C; expected amplifier size: 228 base pair, bp). Gal 2nd amplification primers: Gal 2F: 5′- TCTGGGACTTGGGATGCCTG-3′ and Gal 2R: 5′-GGGGCACATCAACACTTCCT-3′ (AT: 54.1 °C; expected amplifier size: 182 bp)[18]. Gapdh 1st amplification primers: GapdhF: 5′-AACTTTGGCATTGTGGAAGG-3′ and GapdhR: 5′-TGTGAGGGAGATGCTCAGTG-3′ (AT: 61.8 °C; expected amplifier size: 600 bp). Gapdh 2nd amplification primers: Gapdh 2F: 5′-ACCCAGAAGACTGTGGATGG-3′ and Gapdh 2R: 5′-CACATTGGGGTGGTAGGAACAC-3′ (AT: 63.2 °C; expected amplifier size: 171 bp). Gad2 1st amplification primers: Gad2F: 5′-CCAAAAGTTCACGGGCGG-3′ and Gad2R: 5′-TCCTCCAGATTTTGCGGTTG-3′ (AT: 52 °C; expected amplifier size: 375 bp). Gad2 2nd amplification primers: Gad2 2F: 5′-CACCTGCGACCAAAAACCCT-3′ and Gad2 2R: 5′-GATTTTGCGGTTGGTCTGCC-3′ (AT: 54.2 °C; expected amplifier size: 248 bp). Gad1 1st amplification primers: Gad1F: 5′-ATGATACTTGGTGTGGCGTAGC-3′ and Gad1R: 5′-GTTTGCTCCTCCCCGTTCTTAG-3′ (AT: 65 °C; expected amplifier size: 253 bp). Gad1 2nd amplification primers: Gad1 2F: 5′-CAATAGCCTGGAAGAGAAGAGTC-3′ and Gad1 2 R: 5′-GTTTGCTCCTCCCCGTTCTTAG G-3′ (AT: 63 °C; expected amplifier size: 177 bp)[18]. First and second amplification reactions were conducted mixing (4 µl of first-strand cDNA template or pre-amplified, 10 µl reaction mix REDExtract-N-Amp PCR ReadyMix; Sigma-Aldrich and 0.4 µl for each primer; 0.2 to 0.4 µM). The final reaction volume was adjusted to 20 µl with nuclease-free water (Ambion, Thermo Fisher Scientific). Products were visualized by staining with ethidium bromide (Invitrogen) and analyzed by electrophoresis in 1.5-2% agarose gels. We

used Primer 3web 4.1.0 software (http://primer3.wi.mit.edu/) to verify the Gal, Gapdh, Gad1, and Gad2 primers. The sequencing of the second amplification samples that contain the Gal, Gapdh, Gad2 and Gad1 amplicons were conducted by Quintara Bioscience (Boston, MA). We verified the sequence results to the Gal, Gapdh, Gad2 and Gad1 sequences (NCBI GenBank) using Genetyx software. Molecular identification was based on the results from the second reaction. We purchased all the custom primers from Sigma-Aldrich, St. Louis, MO.

For scRT-sqPCR, the cDNAs (n = 38 neurons) were produced as described in the section above. We performed TaqMan™ based probe assay for semi-quantitative gene expressions using the 7500 Fast Real-Time PCR (Applied Biosystems; Foster City, CA). We added cDNA template (2.5–3 µl) enzyme master mix (12.5 µl; TaqMan Universal PCR Master Mix, Applied Biosystems) and TaqMan Gene Expression assays for the amplification of Gal and Gapdh cDNAs (1.25 µl; ThermoFisher Scientific). Volumes for the reactions were adjusted to 25 µl with $H_2O$. The Gal cDNA probe was tagged to FAM/MGB and the Gapdh cDNA to VIC/MGB. TaqMan^TM Gene Expression assays for the Gal cDNA (TaqMan Assay Reagents, Gal, 20X, Cat. #4331182) and for Gapdh cDNA (Pre- Developed Gapdh TaqMan Assay Reagents, 20X, Cat. #4352339E). The thermal cycling conditions were as follows: enzyme activation (95 °C; 10 min), followed by DNA denaturation (95 °C; 15 s; 45–50 cycles) and annealing/extension (60 °C; 1 min). Relative gene expression between samples was calculated using the $2^{-Ct}$ method[53,54]. Data were normalized to the Gapdh expression levels. As control group, we used TdTomato labeled VLPO^GABA/Gal neurons. We used values of 50 for non-detects Ct. Each sample was run in duplicates (2 times, in 2 separate reactions). We considered scRT-PCR- and scRT- sqPCR-negative, samples in which Gapdh expression was undetectable. We represented data as mean ± SEM and n refers to the number of cells per group and we compared group means using Mann-Whitney unpaired t-test.

**Single cell RNA sequencing data analysis**. Single cell RNA-sequencing (scRNA-seq) data of the POA[24] were retrieved from the Gene Expression Omnibus (GEO) repository (GSE113576). Data were available as unique Digital Gene Expression (DGE) matrix. Briefly, the DGE matrix was imported into R software (v.3.6.3) and all the downstream processing was performed using functionalities provided by the R library Seurat (Seurat v.3.1.5)[55–57]. Expression data were filtered according to the following criteria: (i) cells expressing >200 genes and with a mitochondrial gene expression rate <10% were retained; (ii) genes detected in >2 cells were retained. Seurat3 was used to cluster all POA cells and to perform Differential Gene Expression between clusters. Genes expressed only or expressed predominantly in a specific cluster were identified as marker genes. (Supplementary Fig. 9, Supplementary Table 2). Afterwards, clusters corresponding to glial cell types were discarded and neuronal clusters were retained for downstream analyses. According to the spatial information gathered from Multiplexed Error-Robust Fluorescence In Situ Hybridization (Multiplexed Error-Robust Fluorescence In Situ Hybridization)[24] and in situ hybridization from the Allen Brain Atlas, we isolated only neuronal clusters located in the VLPO (#i5, i8, i10, i20, i26, i29, i35, i37, i39, e3, e7, e11, e20 and e24). Briefly, the following Seurat v.3.1.5 functions were used: (1) SCTransform () to normalize and scale the data, to identify the top 3000 variable genes, and to correct for number of unique molecular identifiers (UMIs), percentage of mitochondrial gene expression, and difference between S and G2M cell cycle scores. Cell cycle scores were inferred using CellCycleScoring () function[58]; (2) runPCA was used to calculate the first 50 Principal Components (PCs) of the 3000 most variable features; (3) FindNeighbors () to construct a Shared Nearest Neighbor (SNN) Graph. This method first determines the k-nearest neighbors (knn) of each cell and then uses this knn graph to construct the SNN graph by calculating the neighborhood overlap (Jaccard index) between every cell and its k.param nearest neighbors; (4) FindClusters to define the cell clusters and their granularity using the original Louvain algorithm at resolution of 0.6 for all cells and 1.2 for neurons; (5) runTSNE for t-Distributed Stochastic Neighbor Embedding (t-SNE) dimensional reduction method, to visualize cell clusters in 2 dimensions. (6) FindAllMarkers to perform a differential gene expression between each cluster versus all the other cells of the dataset using the non-parametric Wilcoxon Rank Sum statistics. A gene was defined as differentially expressed if absolute logFC was >0.25 and adjusted p-value (adj-p) (Bonferroni corrected) was <0.05. (7) finally, we used FindMarkers to compare two different cell clusters/groups. The abovementioned parameters were also applied to this function. Cells and neuronal clustering were performed using the same bioinformatic pipeline.

**Statistics and Reproducibility**. Figure 1a: an example of Ox innervation of the POA and VLPO by immunolabeling for Ox-A. The immunolabeling was conducted in n = 4 WT mice. Figure 1c an example of immunolabeling for ChR2-mCherry of n = 5 Ox-IRES-Cre mice used for whole-animal optogenetic studies. Figure 1d: an example of double labelling for ChR2-mCherry and Ox-A immunoreactivity to test selectivity of the AAV-DIO-ChR2-eYFP (and AAV-DIO-ChR2-mCherry) for Ox neurons. Immunolabeling conducted in n = 4 Ox-IRES-Cre mice injected with AAV-DIO-ChR2-eYFP (and AAV-DIO-ChR2-mCherry) into the Ox field. Figure 2a and b: map of n = 29 recorded neurons distributed in n = 19 slices from n = 14 WT mice. Figure 4a: an example of VLPO^GABA neurons (in green) and VLPO^GABA/Gal neurons (double-labeled in red and green) following the injection of a mix of AAV-fDIO-ChR2-eYFP and AAV-DIO-TdTomato into the

VLPO of *Vgat-Flp::Gal-IRES-Cre* mice. This experiment was conducted in *n* = 10 *Vgat-Flp::Gal-IRES-Cre* mice. Figure 5e: an example of triple staining for *Gal*, *Vgat* (*Slc32a1*) and *Ox₂R* mRNAs by RNA scope in situ hybridization. This staining was conducted in *n* = 3 WT mice. Supplementary Fig. 4b: an example of double labeling for biocytin and TdTomato. This labelling was conducted in *n* = 3 recorded neurons distributed in *n* = 2 slices from *n* = 2 *Vgat-IRES-Cre* mice. Supplementary Fig. 5: an example of double labelling for TdTomato (and *AAV-DIO-GFP*) immunolabeling and *Vgat* mRNA in *Vgat-IRES-Cre* mice injected with *AAV-DIO-TdTomato* (and *AAV-DIO-GFP*) into the VLPO. This labelling was conducted in *n* = 4 *Vgat-IRES-Cre* mice. Supplementary Fig. 6b: an example of double labeling for biocytin and TdTomato. This labelling was conducted in *n* = 3 recorded neurons distributed in *n* = 3 slices from *n* = 2 *Gal-IRES-Cre* mice. Supplementary Fig. 6c: an example of recordings from a TdTomato labeled VLPO^GABA/Gal neurons. For these experiments we recorded from *n* = 49 labeled VLPO^GABA/Gal neurons in *n* = 23 slices from *n* = 21 *Gal-IRES-Cre* mice. Supplementary Fig. 7a: an example of recordings from a TdTomato labeled VLPO^GABA/Gal neurons. For these experiments we recorded from *n* = 9 labeled VLPO^GABA/Gal neurons in *n* = 9 slices from *n* = 7 *Gal-IRES-Cre* mice. Supplementary Fig. 8a: an example of double labelling for ChR2-eYFP (immunolabeling) and *Vgat* mRNA in *Vgat-Flp* mice injected with *AAV-fDIO-ChR2-eYFP* into the VLPO. This labelling was conducted in *n* = 2 *Vgat-Flp* mice. Fig. 8b: an example of double labelling for TdTomato (immunolabeling) and *Gal* mRNA in *Gal-IRES-Cre* mice injected with *AAV-DIO-TdTomato* into the VLPO. This labelling was conducted in *n* = 2 *Gal-IRES-Cre* mice.

**Reporting summary**. Further information on research design is available in the Nature Research Reporting Summary linked to this article.

## Data availability
The data generated in this study are presented within this paper or its supplementary materials as well as source data are provided with this paper. These include all individual data points and average values showed in both figures and supplementary information. The raw data for in vivo optogenetic stimulation, in vitro electrophysiology and imaging experiments are available from the corresponding authors, upon reasonable request. For scRNA-seq analysis we provided expression count matrix, barcodes and gene IDs for POA and VLPO and figures at the following link: https://doi.org/10.5281/zenodo. 6570978 as source data. For scRNA-seq raw data, the access code on the GEO repository is GSE113576. Source data are provided with this paper.

## Code availability
All custom codes used in this manuscript are available at the following repository: https://doi.org/10.5281/zenodo.6570978 (version 3) or by the corresponding authors upon request. Other custom codes used in this study were previously published and are available in the relevant citations.

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

## Acknowledgements

This study was supported by NIH grants: R01 NS122589 and P01HL149630 to E.A. R01 NS103161 and R01 NS073613 to P.M.F. R01 NS107315, R01 DK108797, R21 HD098056, P30 DK046200 to D.K. R01 DK075632, R01 DK089044, R01 DK096010, R01 DK122976, P30 DK046200 and P30 DK057521 to B.B.L. R01 NS072337 and R01 NS085477 to C.B.S. The authors thank Dr. Ramalingam Vetrivelan for providing the *Gal-IRES-Cre* mice. We thank Dr. Victoria Petkova, Francesca Raffin and Somdeb Banerjee for their technical assistance. We also thank Dr. Roberto Negro for the help in generating Fig. 7 of this article through BioRender.

## Author contributions

R.D.L., E.A. and P.M.F. designed research; R.D.L., A.V., M.C., S.S.B., L.T.S., B.V. and L.Z., performed research; R.D.L., S.N., K.P.G., A.V., A.Z. and E.A. analyzed data; D.K., T.M., B.B.L., T.E.S. and P.M.F. contributed to unpublished reagents/analytic tools; D.K., B.B.L., P.M.F. and E.A. contributed to the project administration and funding acquisition; C.B.S. contributed as scientific advisor; R.D.L. P.M.F. and E.A. wrote the paper. A.V. and M.C. equally contributed to this study.

## Competing interests

The authors declare no competing interests.
