## [Peer Review File · Nature Communications]

REVIEWER COMMENTS

Reviewer #1 (Remarks to the Author):

Orexin has clear sleep-suppressive effects, yet the mechanisms/circuit through which orexin acts to inhibit sleep have not been defined. It is known that orexin induces wake, that orexin delivered to the VLPO induces wake, and that orexinergic neurons release glutamate in addition to orexin. Both glutamate and orexin are both excitatory, so how the excitatory neuromodulator/transmitter could inhibit sleep has been a puzzle. The present work is an elegant and certainly a most robust study of orexin VLPO circuitry. The authors clearly establish that there are two populations of GABAergic neurons in VLPO (VLPOGABA and VLPOGABA/Gal). They show that orexin subtype 2 receptors (OX2R) are found only on the VLPOGABA neurons, which Orexin excites. In contrast, VLPOGABA/Gal neurons do not have OX2R and are inhibited by Orexin in the VLPO. This sets up an orexin excitation of VLPOGABA that then inhibits VLPOGABA/Gal. This is not a surprising circuit, given the background info leading up to this study, but the work is a fantastic example of how robust circuit mapping should be done.

Nothing imperative to change but the following points should be addressed:

This orexin-VLPO circuit should not be interpreted as “the orexin wake circuit”, and the authors acknowledge this in the discussion, noting that other wake-activated neuromodulators may share the circuit and the activation of wake neurons may circumvent the orexin-VLPO circuit. In essence, the authors show sufficiency without showing that this pathway is essential for wakefulness. Inhibition of the VLPO orexin projections while exciting orexinergic neurons would be needed to discern necessity.

Additionally, orexin, should have unique effects in the VLPO, relative to noradrenaline (NA) and other neuromodulators, as orexin contributes to state (wake and sleep) stabilization. The authors compare the response with the VLPO response to NA, but a contrast would be more insightful. It would be helpful to know whether stimulation of orexinergic neurons results in a more prolonged wake by resulting in a longer activation of the VLPOGABA, relative to stimulation of locus coeruleus neurons. They should be able to comment on duration of wake after optogenetic stim. They may also have data from slice recordings and it should be determined whether orexin’s effects at VLPOGABA neurons are more prolonged than with NA.

Having established the VLPOGABA neurons as a key switch for sleep/wake, and having examined the genetic make up of VLPO, the authors also are super close to a “Holy Grail” in sleep. At present, there are no widely effective and safe hypnotics. If this one group of VLPOGABA neurons has any unique receptors, this would be the perfect group to inhibit and induce sleep. Sleep could then be more effectively induced in stressed individuals with high noradrenaline tone, shift workers with orexin at the wrong time, etc. The authors do show that at least 3 receptors differ for VLPOGABA and VLPOGABA/Gal, where only VLPOGABA neurons have the Ox2R, calcitonin and RORB receptors. Looking for orphan GPCR's unique to these neurons would be an impactful future direction.

Additionally, orexin is more complicated than a switch into wake, as orexin levels have complex relationships with behavioral state. Although orexin levels are higher in wake than in NREMS, they are also higher in REMS than in NREMS and within wakefulness, they are higher in wake with positive emotions (a social encounter) relative to wake with a negative encounter (doctor walking into patient's room). Additionally, the levels across a 24hr period follow an interesting pattern of highest levels (in nocturnal rodents) at lights on (beginning of sleep period) and lowest levels at the beginning of the lights off (active period). Within the dark period levels are lowest during the greatest waking (first two hours), and may follow more of a homeostatic pattern, high with high homeostatic drive and low after greater sleep. All of this is to say that there is still much to learn regarding orexin control of behavioral state.

Minors:

It should be acknowledged that orexin can also suppress sleep/increase wake by exciting noradrenergic locus coeruleus neurons, as Carter et al PNAS 2013, showed that inhibiting the orexinergic excitation of locus coeruleus neurons prevented wake-induced by optogenetic stimulation of orexinergic neurons.

Please scale time for Fig 3C to match time scale bar length in Figs 2 and 3A, as this may also help highlight prolonged activation with OxA.

Reviewer #2 (Remarks to the Author):

Orexin/hypocretin neurons play a major role in wakefulness generation and maintenance, and impaired orexin signaling (mainly by Ox cell loss) produces the sleep disorder narcolepsy. Current circuit models, supported by ample empirical data, suggest that orexins stabilize wakefulness by direct excitatory inputs onto other arousal-promoting systems. While intra-VLPO administration of orexins has been shown to increase wakefulness, suggesting a potential effect on sleep-promoting processes within this region, the precise cellular and circuit mechanisms are unknown.

In this study, the authors conducted an impressive array of experiments including optogenetic stimulation, optogenetic-assisted circuit mapping, in-vitro electrophysiology, and transcriptomics, demonstrating that orexin neurons promote arousal by indirect inhibition of sleep promoting (Galanin+) neurons in the ventrolateral preoptic nucleus of the hypothalamus. The studies were carefully designed and executed, the data are compelling and clearly presented, the manuscript is very well written, and the conclusions are fully supported by the data. The present work is thus novel, highly relevant, scientifically solid, and exciting.

After a careful review of the manuscript (twice) I do not have any meaningful criticisms. My only suggestion to the authors would be to add a set of experiments testing whether inhibition of Ox terminals within the VLPO can produce any narcolepsy-like effects at the behavioral or cortical level. However, these experiments are not critical, nor needed to support the authors' conclusion that Ox neurons increase wakefulness by inhibition of sleep-promoting VLPO neurons.

Last, there is a typo on Page 18, line 540: "...REM sleep "wake" for at least 20s..."

Reviewer #3 (Remarks to the Author):

My sincere complements! I believe this is the first paper I have reviewed in my 50+ years of reviewing that I would say is as close to perfect as you could get. Your work will be a text-book case of how to investigate and describe a neural control circuit. Next challenge is to discover and incorporate the essential feedback loops to make it a regulatory circuit. The text is clearly written, the figures are beautiful, and the conclusions are straight-forward and solidly based on the data. Most significant is that you answer a conundrum in narcolepsy -- why without the arousal stimulus, you still have normal sleep amounts, but fragmented.

Your provocative suggestion at the end that disruptions of this circuit could be involved in parasomnias leads me to wonder what changes/deficits in the circuit could lead to the diversity of parasomnias. This would make an interesting hypothesis paper.

I could mention some minor editorial comments such as why use both orexin and hypocretin terms, and some sentences that could be clearer, but in light of the overall quality of the manuscript they are inconsequential. Craig Heller

Senior Editor & the three Reviewers

We thank all three Reviewers for their positive and supportive comments. During the manuscript revision process we have endeavored to address all of the critiques in full. All changes to the manuscript text have been highlighted in red to facilitate a rapid re-review.

Reviewer #1

This is not a surprising circuit, given the background info leading up to this study, but the work is a fantastic example of how robust circuit mapping should be done.

Nothing imperative to change but the following points should be addressed:

This orexin-VLPO circuit should not be interpreted as “the orexin wake circuit”, and the authors acknowledge this in the discussion, noting that other wake-activated neuromodulators may share the circuit and the activation of wake neurons may circumvent the orexin-VLPO circuit. In essence, the authors show sufficiency without showing that this pathway is essential for wakefulness.

We agree with the reviewer, and as was acknowledged, we did address this issue in our original discussion. We know from our work and work by many others that there is considerable redundancy in the brain's arousal circuitry, with only a few 'nodes' having been established as truly necessary for arousal maintenance. We further agree with the reviewer that orexin neurons likely produce arousal - and are of course necessary for proper state stabilization - through multiple projections and that likely many of the post-synaptic 'targets' of orexin neurons contribute to arousal control, but that each 'target' is unlikely completely necessary for arousal maintenance. Our revised manuscript includes a more expansive discussion of the reviewer's excellent point(s) on page 14. Testing this hypothesis, which we fully agree is an important future direction, is however simply beyond the scope of our current project, which we continue to feel represents a substantial literature contribution unto itself.

Inhibition of the VLPO orexin projections while exciting orexinergic neurons would be needed to discern necessity.

Please see our foregoing comments.

Additionally, orexin, should have unique effects in the VLPO, relative to noradrenaline (NA) and other neuromodulators, as orexin contributes to state (wake and sleep) stabilization.

We agree with the reviewer that the orexin neurons appear to be uniquely positioned to ensure behavioral state stabilization as loss of orexin neurons produces narcolepsy (sleep and wake fragmentation) whereas selective lesions of the locus coeruleus NA neurons produces only small (and insignificant) effects on the amount, timing or consolidation of wakefulness (Gompf et al 2010).

It would be helpful to know whether stimulation of orexinergic neurons results in a more prolonged wake by resulting in a longer activation of the VLPO^{GABA}, relative to stimulation of locus coeruleus neurons. They should be able to comment on duration of wake after optogenetic stim.

The reviewer makes a good point, but we would note that it is always difficult to interpret differences in results obtained by studies that use different stimulation paradigms. We found for example that optogenetic stimulation of orexin input within the VLPO rapidly and reliably rouses mice from both NREM and REM sleep, and that the mice did not immediately go back to sleep following the stimulation events. In fact, the opto-evoked arousals were of similar duration to spontaneous arousals. Carter and colleagues (PNAS 2012) as another excellent example, produced arousals in mice when stimulating orexin input to the LC, but unfortunately the authors did not report the length of the opto-evoked wake bouts, making a direct comparison difficult, if impossible.

They may also have data from slice recordings and it should be determined whether orexin's effects at VLPO^{GABA} neurons are more prolonged than with NA.

We did not find that the response to Ox was longer than that to NA.

Having established the VLPO^{GABA} neurons as a key switch for sleep/wake, and having examined the genetic make up of VLPO, the authors also are super close to a "Holy Grail" in sleep. At present, there are no widely effective and safe hypnotics. If this one group of VLPO^{GABA} neurons has any unique receptors, this would be the perfect group to inhibit and induce sleep.

We could not agree more with the reviewer's comment: the work described represents a major and on-going experimental goal of our labs.

The authors do show that at least 3 receptors differ for VLPO^{GABA} and VLPO^{GABA/Gal}, where only VLPO^{GABA} neurons have the Ox2R, calcitonin and RORB receptors. Looking for orphan GPCR's unique to these neurons would be an impactful future direction.

We fully agree with the reviewer that the expressions of known and orphan GPCRs in VLPO^{GABA} neurons might help with the identification of unique molecular and cellular VLPO 'targets' for against which newer and more selective drugs can be developed.

Additionally, orexin is more complicated than a switch into wake, as orexin levels have complex relationships with behavioral state. Although orexin levels are higher in wake than in NREMS, they are also higher in REMS than in NREMS and within wakefulness, they are higher in wake with positive emotions (a social encounter) relative to wake with a negative encounter (doctor walking into patient's room). Additionally, the levels across a 24hr period follow an interesting pattern of highest levels (in nocturnal rodents) at lights on (beginning of sleep period) and lowest levels at the beginning of the lights off (active period). Within the dark period levels are lowest during the greatest waking (first two hours), and may follow more of a homeostatic pattern, high with high homeostatic drive and low after greater sleep. All of this is to say that there is still much to learn regarding orexin control of behavioral state.

We fully agree with the reviewer's statement and hope to make continued contributions to a more complete understanding of the neurobiology of the Ox system.

Minors:

It should be acknowledged that orexin can also suppress sleep/increase wake by exciting noradrenergic locus coeruleus neurons, as Carter et al PNAS 2013, showed that inhibiting the orexinergic excitation of locus coeruleus neurons prevented wake-induced by optogenetic stimulation of orexinergic neurons.

The reviewer's suggestion is excellent, and we now discuss Carter's important work on page 4.

Please scale time for Fig 3C to match time scale bar length in Figs 2 and 3A, as this may also help highlight prolonged activation with OxA.

We have made the suggested changes. We redrew the scale bars to be identical in every figure.

Reviewer #2

In this study, the authors conducted an impressive array of experiments... The studies were carefully designed and executed, the data are compelling and clearly presented, the manuscript is very well written, and the conclusions are fully supported by the data. The present work is thus novel, highly relevant, scientifically solid, and exciting.

I do not have any meaningful criticisms.

My only suggestion to the authors would be to add a set of experiments testing whether inhibition of Ox terminals within the VLPO can produce any narcolepsy-like effects at the behavioral or cortical level. **However, these experiments are not critical, nor needed to support the authors' conclusion** that Ox neurons increase wakefulness by inhibition of sleep-promoting VLPO neurons.

We thank the reviewer for the suggestion, but as the reviewer acknowledges, testing whether inhibition of orexin input to the VLPO can produce narcolepsy-like effects is outside the scope of the current study. It is an interesting experiment that we intend to pursue, along with the focal (within the VLPO) manipulation of the Ox2R and GABA signaling.

Typo on Page 18, line 540: "...REM sleep "wake" for at least 20s..."

Thank you for the eagle eye edit. We have made this correction.

Reviewer #3

My sincere complements! I believe this is the first paper I have reviewed in my 50+ years of reviewing that I would say is as close to perfect as you could get. Your work will be a text-book case of how to investigate and describe a neural control circuit. Next challenge is to discover and incorporate the essential feedback loops to make it a regulatory circuit. The text is clearly written, the figures are beautiful, and the conclusions are straight-forward and solidly based on the data.

We thank the reviewer (Prof. CH) for his kind and supportive words. We also agree that a more complete understanding of the functional regulatory circuit is an important next step, but also one that will likely take several labs and many years of effort to fully sort out!

REVIEWER COMMENTS

Reviewer #1 (Remarks to the Author):

Orexin has clear sleep-suppressive effects, yet the mechanisms/circuit through which orexin acts to inhibit sleep have not been defined. It is known that orexin induces wake, that orexin delivered to the VLPO induces wake, and that orexinergic neurons release glutamate in addition to orexin. Both glutamate and orexin are both excitatory, so how the excitatory neuromodulator/transmitter could inhibit sleep has been a puzzle. The present work is an elegant and certainly a most robust study of orexin VLPO circuitry. The authors clearly establish that there are two populations of GABAergic neurons in VLPO (VLPOGABA and VLPOGABA/Gal). They show that orexin subtype 2 receptors (OX2R) are found only on the VLPOGABA neurons, which Orexin excites. In contrast, VLPOGABA/Gal neurons do not have OX2R and are inhibited by Orexin in the VLPO. This sets up an orexin excitation of VLPOGABA that then inhibits VLPOGABA/Gal. This is not a surprising circuit, given the background info leading up to this study, but the work is a fantastic example of how robust circuit mapping should be done.

Nothing imperative to change but the following points should be addressed:

This orexin-VLPO circuit should not be interpreted as “the orexin wake circuit”, and the authors acknowledge this in the discussion, noting that other wake-activated neuromodulators may share the circuit and the activation of wake neurons may circumvent the orexin-VLPO circuit. In essence, the authors show sufficiency without showing that this pathway is essential for wakefulness. Inhibition of the VLPO orexin projections while exciting orexinergic neurons would be needed to discern necessity.

Additionally, orexin, should have unique effects in the VLPO, relative to noradrenaline (NA) and other neuromodulators, as orexin contributes to state (wake and sleep) stabilization. The authors compare the response with the VLPO response to NA, but a contrast would be more insightful. It would be helpful to know whether stimulation of orexinergic neurons results in a more prolonged wake by resulting in a longer activation of the VLPOGABA, relative to stimulation of locus coeruleus neurons. They should be able to comment on duration of wake after optogenetic stim. They may also have data from slice recordings and it should be determined whether orexin's effects at VLPOGABA neurons are more prolonged than with NA.

Having established the VLPOGABA neurons as a key switch for sleep/wake, and having examined the genetic make up of VLPO, the authors also are super close to a “Holy Grail” in sleep. At present, there are no widely effective and safe hypnotics. If this one group of VLPOGABA neurons has any unique receptors, this would be the perfect group to inhibit and induce sleep. Sleep could then be more effectively induced in stressed individuals with high noradrenaline tone, shift workers with orexin at the

wrong time, etc. The author do show that at least 3 receptors differ for VLPOGABA and VLPOGABA/Gal, where only VLPOGABA neurons have the Ox2R, calcitonin and RORB receptors. Looking for orphan GPCR's unique to these neurons would be an impactful future direction.

Additionally, orexin is more complicated than a switch into wake, as orexin levels have complex relationships with behavioral state. Although orexin levels are higher in wake than in NREMS, they are also higher in REMS than in NREMS and within wakefulness, they are higher in wake with positive emotions (a social encounter) relative to wake with a negative encounter (doctor walking into patient's room). Additionally, the levels across a 24hr period follow an interesting pattern of highest levels (in nocturnal rodents) at lights on (beginning of sleep period) and lowest levels at the beginning of the lights off (active period). Within the dark period levels are lowest during the greatest waking (first two hours), and may follow more of a homeostatic pattern, high with high homeostatic drive and low after greater sleep. All of this is to say that there is still much to learn regarding orexin control of behavioral state.

Minors:

It should be acknowledged that orexin can also suppress sleep/increase wake by exciting noradrenergic locus coeruleus neurons, as Carter et al PNAS 2013, showed that inhibiting the orexinergic excitation of locus coeruleus neurons prevented wake-induced by optogenetic stimulation of orexinergic neurons.

Please scale time for Fig 3C to match time scale bar length in Figs 2 and 3A, as this may also help highlight prolonged activation with OxA.

Reviewer #2 (Remarks to the Author):

Orexin/hypocretin neurons play a major role in wakefulness generation and maintenance, and impaired orexin signaling (mainly by Ox cell loss) produces the sleep disorder narcolepsy. Current circuit models, supported by ample empirical data, suggest that orexins stabilize wakefulness by direct excitatory inputs onto other arousal-promoting systems. While intra-VLPO administration of orexins has been shown to

increase wakefulness, suggesting a potential effect on sleep-promoting processes within this region, the precise cellular and circuit mechanisms are unknown.

In this study, the authors conducted an impressive array of experiments including optogenetic stimulation, optogenetic-assisted circuit mapping, in-vitro electrophysiology, and transcriptomics, demonstrating that orexin neurons promote arousal by indirect inhibition of sleep promoting (Galanin+) neurons in the ventrolateral preoptic nucleus of the hypothalamus. The studies were carefully designed and executed, the data are compelling and clearly presented, the manuscript is very well written, and the conclusions are fully supported by the data. The present work is thus novel, highly relevant, scientifically solid, and exciting.

After a careful review of the manuscript (twice) I do not have any meaningful criticisms. My only suggestion to the authors would be to add a set of experiments testing whether inhibition of Ox terminals within the VLPO can produce any narcolepsy-like effects at the behavioral or cortical level. However, these experiments are not critical, nor needed to support the authors' conclusion that Ox neurons increase wakefulness by inhibition of sleep-promoting VLPO neurons.

Last, there is a typo on Page 18, line 540: "...REM sleep "wake" for at least 20s..."

Reviewer #3 (Remarks to the Author):

My sincere complements! I believe this is the first paper I have reviewed in my 50+ years of reviewing that I would say is as close to perfect as you could get. Your work will be a text-book case of how to investigate and describe a neural control circuit. Next challenge is to discover and incorporate the essential feedback loops to make it a regulatory circuit. The text is clearly written, the figures are beautiful, and the conclusions are straight-forward and solidly based on the data. Most significant is that you answer a conundrum in narcolepsy -- why without the arousal stimulus, you still have normal sleep amounts, but fragmented.

Your provocative suggestion at the end that disruptions of this circuit could be involved in parasomnias leads me to wonder what changes/deficits in the circuit could lead to the diversity of parasomnias. This would make an interesting hypothesis paper.

I could mention some minor editorial comments such as why use both orexin and hypocretin terms, and some sentences that could be clearer, but in light of the overall quality of the manuscript they are inconsequential. Craig Heller

Senior Editor & the three Reviewers

We thank all three Reviewers for their positive and supportive comments. During the manuscript revision process we have endeavored to address all of the critiques in full.

Reviewer #1

This is not a surprising circuit, given the background info leading up to this study, but the work is a fantastic example of how robust circuit mapping should be done.

Nothing imperative to change but the following points should be addressed:

This orexin-VLPO circuit should not be interpreted as “the orexin wake circuit”, and the authors acknowledge this in the discussion, noting that other wake-activated neuromodulators may share the circuit and the activation of wake neurons may circumvent the orexin-VLPO circuit. In essence, the authors show sufficiency without showing that this pathway is essential for wakefulness.

We agree with the reviewer, and as was acknowledged, we did address this issue in our original discussion. We know from our work and work by many others that there is considerable redundancy in the brain's arousal circuitry, with only a few 'nodes' having been established as truly necessary for arousal maintenance. We further agree with the reviewer that orexin neurons likely produce arousal - and are of course necessary for proper state stabilization - through multiple projections and that likely many of the post-synaptic 'targets' of orexin neurons contribute to arousal control, but that each 'target' is unlikely completely necessary for arousal maintenance. Our revised manuscript includes a more expansive discussion of the reviewer's excellent point(s) on page 14. Testing this hypothesis, which we fully agree is an important future direction, is however simply beyond the scope of our current project, which we continue to feel represents a substantial literature contribution unto itself.

Inhibition of the VLPO orexin projections while exciting orexinergic neurons would be needed to discern necessity.

Please see our foregoing comments.

Additionally, orexin, should have unique effects in the VLPO, relative to noradrenaline (NA) and other neuromodulators, as orexin contributes to state (wake and sleep) stabilization.

We agree with the reviewer that the orexin neurons appear to be uniquely positioned to ensure behavioral state stabilization as loss of orexin neurons produces narcolepsy (sleep and wake fragmentation) whereas selective lesions of the locus coeruleus NA neurons produces only small (and insignificant) effects on the amount, timing or consolidation of wakefulness (Gompf et al 2010).

It would be helpful to know whether stimulation of orexinergic neurons results in a more prolonged wake by resulting in a longer activation of the VLPO^{GABA}, relative to stimulation of locus coeruleus neurons. They should be able to comment on duration of wake after optogenetic stim.

The reviewer makes a good point, but we would note that it is always difficult to interpret differences in results obtained by studies that use different stimulation paradigms. We found for

example that optogenetic stimulation of orexin input within the VLPO rapidly and reliably rouses mice from both NREM and REM sleep, and that the mice did not immediately go back to sleep following the stimulation events. In fact, the opto-evoked arousals were of similar duration to spontaneous arousals. Carter and colleagues (PNAS 2012) as another excellent example, produced arousals in mice when stimulating orexin input to the LC, but unfortunately the authors did not report the length of the opto-evoked wake bouts, making a direct comparison difficult, if impossible.

They may also have data from slice recordings and it should be determined whether orexin's effects at VLPO^{GABA} neurons are more prolonged than with NA.

We did not find that the response to Ox was longer than that to NA.

Having established the VLPO^{GABA} neurons as a key switch for sleep/wake, and having examined the genetic make up of VLPO, the authors also are super close to a "Holy Grail" in sleep. At present, there are no widely effective and safe hypnotics. If this one group of VLPO^{GABA} neurons has any unique receptors, this would be the perfect group to inhibit and induce sleep.

We could not agree more with the reviewer's comment: the work described represents a major and on-going experimental goal of our labs.

The authors do show that at least 3 receptors differ for VLPO^{GABA} and VLPO^{GABA/Gal}, where only VLPO^{GABA} neurons have the Ox2R, calcitonin and RORB receptors. Looking for orphan GPCR's unique to these neurons would be an impactful future direction.

We fully agree with the reviewer that the expressions of known and orphan GPCRs in VLPO^{GABA} neurons might help with the identification of unique molecular and cellular VLPO 'targets' for against which newer and more selective drugs can be developed.

Additionally, orexin is more complicated than a switch into wake, as orexin levels have complex relationships with behavioral state. Although orexin levels are higher in wake than in NREMS, they are also higher in REMS than in NREMS and within wakefulness, they are higher in wake with positive emotions (a social encounter) relative to wake with a negative encounter (doctor walking into patient's room). Additionally, the levels across a 24hr period follow an interesting pattern of highest levels (in nocturnal rodents) at lights on (beginning of sleep period) and lowest levels at the beginning of the lights off (active period). Within the dark period levels are lowest during the greatest waking (first two hours), and may follow more of a homeostatic pattern, high with high homeostatic drive and low after greater sleep. All of this is to say that there is still much to learn regarding orexin control of behavioral state.

We fully agree with the reviewer's statement and hope to make continued contributions to a more complete understanding of the neurobiology of the Ox system.

Minors:

It should be acknowledged that orexin can also suppress sleep/increase wake by exciting noradrenergic locus coeruleus neurons, as Carter et al PNAS 2013, showed that inhibiting the orexinergic excitation of locus coeruleus neurons prevented wake-induced by optogenetic stimulation of orexinergic neurons.

The reviewer's suggestion is excellent, and we now discuss Carter's important work on page 4.

Please scale time for Fig 3C to match time scale bar length in Figs 2 and 3A, as this may also help highlight prolonged activation with OxA.

We have made the suggested changes. We redrew the scale bars to be identical in every figure.

Reviewer #2

In this study, the authors conducted an impressive array of experiments... The studies were carefully designed and executed, the data are compelling and clearly presented, the manuscript is very well written, and the conclusions are fully supported by the data. The present work is thus novel, highly relevant, scientifically solid, and exciting.

I do not have any meaningful criticisms.

My only suggestion to the authors would be to add a set of experiments testing whether inhibition of Ox terminals within the VLPO can produce any narcolepsy-like effects at the behavioral or cortical level. However, these experiments are not critical, nor needed to support the authors' conclusion that Ox neurons increase wakefulness by inhibition of sleep-promoting VLPO neurons.

We thank the reviewer for the suggestion, but as the reviewer acknowledges, testing whether inhibition of orexin input to the VLPO can produce narcolepsy-like effects is outside the scope of the current study. It is an interesting experiment that we intend to pursue, along with the focal (within the VLPO) manipulation of the Ox2R and GABA signaling.

Typo on Page 18, line 540: "...REM sleep "wake" for at least 20s..."

Thank you for the eagle eye edit. We have made this correction.

Reviewer #3

My sincere complements! I believe this is the first paper I have reviewed in my 50+ years of reviewing that I would say is as close to perfect as you could get. Your work will be a text-book case of how to investigate and describe a neural control circuit. Next challenge is to discover and incorporate the essential feedback loops to make it a regulatory circuit. The text is clearly written, the figures are beautiful, and the conclusions are straight-forward and solidly based on the data.

We thank the reviewer (Prof. CH) for his kind and supportive words. We also agree that a more complete understanding of the functional regulatory circuit is an important next step, but also one that will likely take several labs and many years of effort to fully sort out!